# Curriculum Model Merging: Harmonizing Chemical LLMs for Enhanced Cross-Task Generalization

**Baoyi He**[*]
Zhejiang University
12321037@zju.edu.cn

**Luotian Yuan**[*]
Zhejiang University
12221240@zju.edu.cn

**Ying Wei**[†]
Zhejiang University
ying.wei@zju.edu.cn

**Fei Wu**[†]
Zhejiang University
wufei@zju.edu.cn

## Abstract

The emergence of large language models (LLMs) has opened new opportunities for AI-driven chemical problem solving. However, existing chemical LLMs are typically tailored to specific task formats or narrow domains, limiting their capacity to integrate knowledge and generalize across tasks. Model merging offers a promising route for efficiently combining specialized LLMs into a unified model without access to original training data, which is urgently needed in the chemical domain where in-house data and privacy preservation are critical. However, effective model merging in the chemical domain poses unique challenges: (1) significant disparities among chemical LLMs due to task-specific specialization, and (2) a highly imbalanced distribution of chemical LLMs in targeted downstream tasks, where some are over-benchmarked while others remain underexplored. These challenges intensify model inconsistencies such as parameter interference and accumulated fine-tuning noise, which collectively hinder effective model merging. To this end, we propose Curriculum Model Merging (CMM), a curriculum-based framework that progressively merges expert chemical LLMs in a moderate and continual manner. CMM aims to harmonize their inconsistencies while meantime preserve their domain-specific expertise. Comprehensive experiments on two benchmark datasets show that CMM effectively consolidates task-specific expertise and outperforms the state-of-the-art methods by 29.03% in terms of overall average performance. Moreover, CMM facilitates chemical knowledge generalization across prediction and generative tasks without sacrificing robustness, exhibiting promising merging performance under both expert-abundant and expert-sparse scenarios.

## 1 Introduction

The emergence of large language models (LLMs) has profoundly reshaped the landscape of Chemistry [19], demonstrating remarkable effectiveness across a wide spectrum of real-world problems (e.g., property prediction [49], molecular generation [47], and retrosynthesis [15]). A prevailing strategy to adapt foundation LLMs for chemical research is *task-specific fine-tuning*, which enhances downstream performance on specialized datasets. However, this paradigm inevitably fragments chemical intelligence, as each fine-tuned model becomes confined to narrow domains or data formats, limiting holistic knowledge integration and cross-task generalization. *Multi-task learning* partially alleviates the issues by training a universal model on multiple chemical datasets. Yet, its applicability

---

[*]Equal contribution.
[†]Corresponding author.

39th Conference on Neural Information Processing Systems (NeurIPS 2025).

in chemistry domains remains constrained by the scarcity of publicly shareable data under in-house data privacy regulations and the high computation cost required for large-scale joint optimization [62].

Recently, *model merging* has emerged as a compelling alternative for integrating multiple *expert models* into a comprehensive *base model* without requiring access to the original training data or extensive retraining, an approach that has already achieved considerable success in natural language processing (NLP) [57]. However, transferring such success to the chemical domain remains highly challenging. First, there exist significant disparities among expert chemical LLMs. It is widely acknowledged that chemical tasks differ fundamentally from natural language tasks, often requiring larger shifts in representation space and greater parameter adjustments when fine-tuning general-purpose foundation models (e.g., LLaMA [46], GPT [40]) for chemical applications. Consequently, base and expert models in the chemical domain tend to exhibit greater variance due to task-specific specializations, in stark contrast to the relatively homogeneous model landscape in NLP. Additionally, these greater fine-tuning adjustments accumulate noise and further amplify parameter inconsistencies, leading to interference and hidden conflicts among expert models that complicate the merging process. Second, the chemical domain exhibits a highly imbalanced distribution of targeted downstream tasks. Widely studied tasks such as property prediction are supported by numerous high-performing models (i.e., expert-abundant scenarios), whereas many niche or emerging tasks such as retrosynthesis are covered by only a few or no published models (i.e., expert-sparse scenarios). This imbalanced distribution introduces additional challenges when merging models: knowledge critical to expert-sparse tasks may be diluted by noise or overshadowed by dominant patterns from expert-abundant models. As a result, the effectiveness of existing model merging methods remains limited in chemical domain. Merged models frequently suffer from poor knowledge generalization beyond individual model boundaries and degraded performance across diverse task types.

In this paper, we introduce Curriculum Model Merging (CMM), a framework designed to address the aforementioned challenges in merging chemical LLMs. Inspired by curriculum learning [4], we decompose the extremely complicated problem of simultaneously combining heterogeneous expert models into a progressive and continual merging process. At its core, CMM constructs a route-based merging curriculum. In the early stages, weaker expert models are merged first, which gradually enhances the base model's generalization and stability. This, in turn, prepares the base model to accommodate stronger and more specialized experts in subsequent rounds. The result is a merged model that incrementally strengthens task-specific capabilities while maintaining robustness across tasks. Our CMM method consists of two main steps. (1) *Curriculum construction*: CMM ranks expert models based on their capabilities across various benchmarks, establishing an ordered curriculum that navigates a continual, capability-aware merging process. (2) *Iterative merging*: each expert model is progressively merged into the current base model through task vector extraction and composition, after which the merged model serves as the base model for the next iteration. Recognizing that the relative importance of each expert significantly influences the ultimate performance, we introduce multiple merging strategies that vary the degree of involvement for each expert. This is achieved by jointly controlling the merging order and assigning adaptive merging weights.

The key contributions of our paper are outlined below. (1) *Practicability:* We, for the first time, integrate model merging techniques to effectively consolidate task-specific specializations in the chemical domain, overcoming data unavailability and avoiding substantial computational costs. (2) *Efficacy:* We introduce CMM, a progressive and performance-aware merging framework that addresses the significant challenges of merging heterogeneous chemical LLMs. Across two representative benchmarks, CMM exceeds the state-of-the-art merging method by 29.03%. (3) *Generality:* We evaluate CMM under both expert-abundant and expert-sparse scenarios. The results demonstrate that CMM facilitates robust chemical knowledge generalization across both predictive and generative tasks, without sacrificing performance consistency.

## 2 Related Work

**Chemical LLMs**    We briefly summarize the recently published LLM studies and their specialized versions in the chemistry domain in Appendix A. Early models such as SMILES-BERT [49] and SMILES Transformer [22] have been employed to process SMILES information of molecules. Uni-Mol [64], AGBT [5] and Molformer [52] focus on modeling 3D molecules. SciGLM [61], SciL-itLLM [30], KALE-LM-Chem [9], and LLAMAT [37] primarily focus on natural language processing, information extraction, and scientific reasoning capabilities. GeLLMO [11], MolecularGPT [32], and

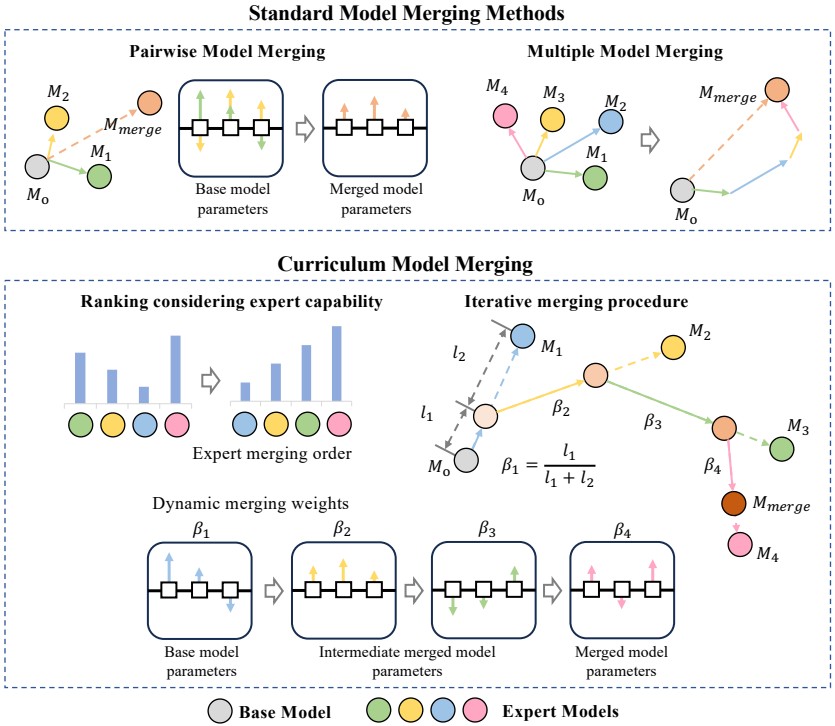

Figure 1: **Standard Model Merging Methods vs. Curriculum Model Merging** Standard Model Merging merges all expert models simultaneously, while Curriculum Model Merging first ranks the expert models and then merges them progressively.

Mol-LLaMA [27] have been fine-tuned for molecule-related tasks, including multi-property molecule optimization tasks, molecular property prediction, and understanding molecules. ChemLLM [62], ChemDFM [63], LlaSMol [59] and Molinst-molecule [15] are specialized for essential chemistry tasks, such as retrosynthesis, yield prediction, molecular property prediction, and description-guided molecule design. CrystaLLM [1] and Crystal-text-LLM [38] are specifically designed for crystal generation and stable materials generation.

**Model Merging Methods**   Model merging methods are broadly categorized into: (1) pre-merging methods, which align models before merging, and (2) during-merging methods, which resolve conflicts during the merging process. Pre-merging methods leverage the linear mode connectivity (LMC) property [12, 14, 16] to align models prior to merging. Early approaches such as CCAMerge [23] use CCA-based neuron alignment, while MuDSC [54] jointly aligns weights and activations. C2M3 [7] optimizes global permutations layer-wise, and Deep-Align [39] introduces a learnable architecture for dynamic weight alignment. During-merging methods directly manipulate parameters to reduce interference. Model Soups [51] averages weights, while Task Arithmetic [25] and TIES [55] use task vectors with interference control. DARE [60] and DELLA [10] sparsify deltas to retain critical changes. Uncertainty-based merging [8] uses second-order Hessians to guide merging. Modular approaches such as Concrete [42], EMR-Merging [24], Twin-Merging [34], and Weight-Ensembling MoE [44] apply dynamic routing and task-specific control. PWE-MoE [43] extends this to multi-objective settings using preference vectors. Finally, Representation Surgery [58] calibrates merged models post hoc by aligning their representations with those of the original expert models.

## 3 Methodology

### 3.1 Preliminaries

Let $M_o$ represents a pretrained model and $\{M_1, M_2, \ldots, M_N\}$ represent models fine-tuned based on the pretrained model. The goal is to merge the combined strengths of these models into $M_o$. We

follow the task arithmetic [25] approach and merge tasks using combined task vectors $\tau_t$, where the task vector is defined as

$$\tau_t = \theta_{\text{ft}}^t - \theta_{\text{pre}}, \tag{1}$$

where $\theta_{\text{pre}}$ represents the weights of the pretrained model and $\theta_{\text{ft}}^t$ represents weights after fine-tuning on task $t$.

**Task Arithmetic Merging**   Given $N$ expert models fine-tuned from the same base, this method aims to merge them into a single model $\theta_{\text{merge}}$ by linearly combining their task vectors. A simple arithmetic average is used to obtain the merged task vector:

$$\tau_{\text{new}} = \sum_{t=1}^{n} \tau_t, \quad \theta_{\text{merge}} = \theta_{\text{pre}} + \lambda * \tau_{\text{new}} \tag{2}$$

where $\lambda$ is an optional scaling term controlling the influence of the aggregated task updates, and $\tau_{\text{new}}$ denotes the sum of all task vectors.

**Drop And Rescale Merging [60]**   This is a method to reduce interference during merging. The first step, Drop, randomly resets a portion of task vectors to zero according to a drop rate $p$. The second step, Rescale, multiplies the remaining values by a scaling factor of $\frac{1}{1-p}$. After applying Drop And Rescale (DARE), a large number of parameters in the fine-tuned model become identical to those in the base model. In other words, DARE updates only a small subset of parameters, retaining only a sparse set of parameter changes and thereby reducing redundancy.

### 3.2   Curriculum Model Merging

We evaluate the set of models $\{M_1, M_2, \ldots, M_N\}$ on a validation set. For each model, we evaluate its performance on individual tasks and normalize the scores to ensure comparability across tasks. The normalized scores are then averaged and aggregated to compute an overall performance score. Let the overall performance scores be denoted as $\{S_1, S_2, \ldots, S_N\}$. Note that we provide an ablation study in Appendix C to investigate the impact of different score computation methods as well as various sizes of the validation set on final merging effectiveness. We sort the models in ascending order of performance to obtain a ranking:

$$M_{(1)}, M_{(2)}, \ldots, M_{(N)}, \quad \text{such that} \quad S_{(1)} \leq S_{(2)} \leq \ldots \leq S_{(N)},$$

where $S_{(i)}$ is the score of model $M_{(i)}$. The coefficient $\beta_k \in (0, 1)$ is the scaling term used to combine the $k$-th task vector. Specifically, we compute:

$$\tau_{(1)} = \theta_{\text{ft}}^{(1)} - \theta_{\text{pre}}, \quad \theta_1 = \theta_{\text{pre}} + \beta_1 \cdot \tau_{(1)}, \tag{3}$$

$$\tau_{(2)} = \theta_{\text{ft}}^{(2)} - \theta_1, \quad \theta_2 = \theta_1 + \beta_2 \cdot \tau_{(2)}, \tag{4}$$

$$\ldots$$

$$\tau_{(k)} = \theta_{\text{ft}}^{(k)} - \theta_{k-1}, \quad \theta_k = \theta_{k-1} + \beta_k \cdot \tau_{(k)}, \tag{5}$$

where $k = 1, 2, \ldots, N$. $\tau_{(k)}$ represents the difference between the fine-tuned parameters of the $k$-th model and the current merged model, and $\theta_k$ denotes the parameters of the merged model after incorporating the $k$-th task vector. This recursive procedure continues until all task vectors have been merged.

Since model performance improves progressively, better-performing models should not be assigned smaller weights. There are two strategies to increase the contribution of the later models. First, the distribution of $\beta$ should be either flat or increasing based on practical needs, such as constant value, linear schedule, or exponential growth. We have explored these possibilities in Section 4.4. Second, CMM is capable of dynamically adjusting the degree of involvement of each expert. Since $0 < \beta_k < 1$, this property inherently allows greater weights for the later models. The proof is provided below.

Table 1: Performance of baseline methods and the proposed approach on Chembench. For all evaluation metrics, higher values indicate better performance. We highlight the best performance as bold. Bold values denote the best results in each column.

| Model | NC | Property_P | M2C | C2M | Product_P | RS | YP | TP | SP | Average |
|---|---|---|---|---|---|---|---|---|---|---|
| **Expert models** | | | | | | | | | | |
| Llama-3-8B-Instruct | 51.19 | 27.79 | 90.30 | 40.88 | 34.00 | 29.33 | 45.33 | 60.89 | 33.67 | 45.93 |
| Molinst-molecule-8b | 39.05 | 25.39 | 80.94 | 39.75 | 29.67 | 31.67 | 46.33 | 60.89 | 33.00 | 42.97 |
| KALE-LM-Chem-1.5 | 61.33 | **43.72** | 90.30 | 53.75 | 72.67 | 53.67 | 45.67 | 47.52 | 45.00 | 57.07 |
| Meerkat-8b-v1.0 | 50.19 | 24.26 | 86.62 | 40.88 | 27.33 | 30.33 | 42.33 | 56.93 | 34.00 | 43.65 |
| GeLLMO-P6 | 28.91 | 29.48 | 59.53 | 24.25 | 24.67 | 25.00 | 41.33 | 29.70 | 24.00 | 31.87 |
| **SOTA merge methods** | | | | | | | | | | |
| TA | 60.83 | 25.11 | **92.98** | 46.12 | 55.00 | 46.67 | **47.00** | 59.90 | 33.00 | 51.85 |
| TIES | 46.43 | 30.61 | 77.26 | 40.00 | 35.67 | 34.33 | 32.33 | 44.55 | 27.33 | 40.95 |
| DARE_TA | 31.41 | 25.11 | 49.83 | 23.88 | 25.00 | 25.00 | 41.67 | 29.21 | 23.33 | 30.49 |
| SCE | 40.05 | 43.02 | 77.93 | 36.75 | 29.00 | 28.67 | 45.33 | 30.20 | 29.33 | 40.03 |
| AF | 58.95 | 29.76 | 91.97 | 47.5 | 47.00 | 42.00 | 46.00 | 63.37 | 37.33 | 51.54 |
| **Merged models through CMM** | | | | | | | | | | |
| CMM | **63.95** | 35.40 | **92.98** | **54.37** | **78.33** | **72.33** | 46.00 | **64.85** | 44.67 | **61.43** |
| DARE_CMM | 63.58 | 33.43 | 92.31 | 51.88 | 74.00 | 68.00 | 47.33 | 62.87 | **46.00** | 59.93 |

Substituting $\tau_{(1)}$ into $\theta_1$ gives:

$$\theta_1 = (1 - \beta_1)\theta_{\text{pre}} + \beta_1\theta_{\text{ft}}^{(1)} \tag{6}$$

Similarly, substituting the previous expressions into $\theta_2$ gives:

$$\theta_2 = (1 - \beta_1)(1 - \beta_2)\theta_{\text{pre}} + \beta_1(1 - \beta_2)\theta_{\text{ft}}^{(1)} + \beta_2\theta_{\text{ft}}^{(2)} \tag{7}$$

$$\cdots$$

$$\theta_k = \left(\prod_{i=1}^{k}(1 - \beta_i)\right)\theta_{\text{pre}} + \sum_{j=1}^{k}\left[\beta_j\left(\prod_{i=j+1}^{k}(1 - \beta_i)\right)\theta_{\text{ft}}^{(j)}\right] \tag{8}$$

During merging, since $0 < \beta_k < 1, 0 < (1 - \beta_k) < 1$, the involvement of top-ranked models that are merged earlier is dynamically reduced.

Our framework is compatible with different and incoming advanced model merging methods. We denote the original version combined with Task Arithmetic as CMM, and the one with DARE as DARE_CMM.

## 4 Experiments

### 4.1 Experimental Setup

**Base & expert models** We briefly summarize the recently published LLM studies in the chemistry domain in Appendix A. Among them, expert models for merging are selected considering their enhanced performance on respective specialized tasks. The performance of each model on individual chemical tasks is presented in Table 1 and Table 2. Ultimately, we adopt the universal LLaMA3-8B-Instruct [18] as the **base model**, GeLLMO-P6-Llama [11], Meerkat-8B-v1.0 [28], KALE-LM-Chem-1.5 [9] and Molinst-Molecule-8B [15] as **expert models**. Based on the method which is demonstrated in Section 3.2 and the overall performance across benchmarks of each expert models reported in Table 3, the **merging order** is assigned as follow: GeLLMO-P6-Llama, Meerkat-8B-v1.0, Molinst-Molecule-8B, and KALE-LM-Chem-1.5. The **merging weight** coefficients for each model, $\beta$, are assigned using a linear strategy that increase from **0.3** to **0.6**. We draw inspiration for this coefficient range from a conclusion in task arithmetic [25]: "Scaling coefficients in the range 0.3 to 0.5 produce close to optimal results in many cases." Ablation studies and corresponding experimental results are provided in Section 4.4 to demonstrate the effectiveness of the chosen merging order and the $\beta$ weighting strategy.

**Baseline algorithms** We compare the performance of the merged model derived from our approach against a variety of baselines, including: 1) the original base and expert models, and 2) merged

models obtained through existing high-performance model merging approaches. Task Arithmetic (**TA**) [25] constructs task vector by subtracting the weights of the base model from the expert models and combines together the task vectors by concise arithmetic operations. Ties-merging (**TIES**) [55] discards the task vectors with negligible changes and combines the remaining vectors that are aligned in sign. **DARE_TA** [60] considers the disparities between base and expert model parameters, dropping a subset of task vectors. **SCE** [48] allocates parameter matrix-level coefficients based on the magnitude of parameter changes, enabling fine-grained merging. Arcee Fusion (**AF**) [17] assesses the importance of each parameter based on the Kullback–Leibler (KL) divergence and dynamically optimizes the parameters accordingly. Studies have shown that across different datasets and experimental settings, **TA**, **DARE_TA**, and **TIES** alternately achieve the best performance and are regarded as state-of-the-art methods. To ensure a fair comparison, all approaches use the same base and expert models described above. While **TA**, **TIES**, **DARE_TA**, and **SCE** support the simultaneous merging of multiple models, **AF** merges one expert model into the base model at a time and performs iterative merging for multiple experts. Each method receives the same ranking information derived from the validation set. More specifically, the TA, TIES, DARE_TA, and SCE methods also require assigning a weight to each expert model. The AF method requires specifying a merging order. In all cases, the weights or orders used are the same as those used by CMM. The details and results of the comparison between CMM and other machine learning methods are provided in Appendix B.

**Benchmarks and evaluation metrics** The performance of various models is evaluated on two representative chemical benchmarks. The first benchmark, Chembench [62], consists of 4,100 high-quality multiple-choice questions and answers spanning 9 core chemistry tasks: Name Conversion (**NC**), Property Prediction (**Property_P**), Mol2Caption (**M2C**), Caption2Mol (**C2M**), Product Prediction (**Product_P**), Retrosynthesis (**RS**), Yield Prediction (**YP**), Temperature Prediction (**TP**), and Solvent Prediction (**SP**). The evaluation metric used for this benchmark is **accuracy**. The second benchmark consists of two molecular generation tasks from Mol-Instructions [15]: retrosynthesis and forward reaction prediction. To reduce computational overhead, 200 samples from each task are randomly selected as the test dataset. An ablation study in Appendix D examines the relationship between the number of evaluation samples and the resulting metrics. The results indicate that beyond 200 samples, the evaluation scores stabilize, showing only minor fluctuations without any consistent upward or downward trend. We employ Round-trip accuracy, SELFIES BLEU score, Validity rate and Exact match rate as the evaluation metrics to comprehensively evaluate the performance. SELF-IES BLEU score (**SBS**) quantifies the syntactic similarity between the generated SELFIES strings and the ground truth strings using the BLEU metric, which captures n-gram overlap. Validity rate (**VR**) is defined as the proportion of generated molecular structures that are chemically valid. Exact match rate (**EMR**) measures the proportion of generated molecular structures that exactly match the corresponding ground truth structures. However, given that multiple chemically plausible predictions may exist beyond the exact ground truth, retrosynthetic studies [26] have proposed a more chemically meaningful and robust evaluation metric, round-trip accuracy. Round-trip accuracy (**RTA**) measures the proportion of predicted reactants that, when input into a forward reaction model, regenerate the original product. It reflects the chemical validity and semantic correctness of the predicted structures. Accordingly, we consider round-trip accuracy as the most reliable metric on the second benchmark. The validation set used for ranking is composed of the ChemBench dev set and 100 samples from Mol-Instructions, both of which are entirely disjoint from the test set.

## 4.2 Results

Table 1 shows the prediction performance of models merged using our approach (CMM, DARE_CMM), in comparison with individual expert models and models produced by baseline merging approaches across 9 multiple-choice chemical question tasks and their overall average on ChemBench. The expert models exhibit diverse strengths across different tasks, confirming their respective specializations. For example, KALE-LM-Chem-1.5 achieves the best performance on Property_P and strong results on C2M and Product_P, while Molinst-Molecule-8B and Meerkat-8B-v1.0 perform competitively in YP and TP. In contrast, the general-purpose base model LLaMA3-8B-Instruct delivers moderate results and lacks task-specific specialization, leading to a lower average accuracy of 45.93 compared to the best-performing expert, KALE-LM-Chem-1.5, which achieves 57.07. This disparity underscores the value of leveraging expert models tailored to specific chemical domains. Our merged models, CMM and DARE_CMM, consistently outperform individual experts

Table 2: Performance of baselines and our approach on Mol-Instructions. For all columns, higher values reflect better performance.

| Models | Retrosynthesis | | | | Forward Reaction Prediction | | | | Average RTA |
|---|---|---|---|---|---|---|---|---|---|
| | RTA | SBS | VR | EMR | RTA | SBS | VR | EMR | RTA |
| **Expert models** | | | | | | | | | |
| Llama-3-8B-Instruct | 0 | 0 | 1.00 | 0 | 0.14 | 0.21 | 0.82 | 0 | 0.07 |
| Molinst-molecule-8b | **0.42** | 0.50 | 0.66 | 0.22 | **0.31** | 0.55 | 0.69 | 0.33 | **0.37** |
| KALE-LM-Chem-1.5 | 0 | 0 | 1.00 | 0 | 0.14 | 0.21 | 0.82 | 0 | 0.07 |
| Meerkat-8b-v1.0 | 0 | 0.02 | 0.61 | 0 | 0 | 0.13 | 0.45 | 0 | 0 |
| GeLLMO-P6 | 0 | 0.08 | 0.96 | 0 | 0.16 | 0.14 | 0.93 | 0 | 0.08 |
| **SOTA merge methods** | | | | | | | | | |
| TA | 0 | 0.80 | 0.69 | 0.01 | 0.16 | 0.84 | 0.98 | 0.03 | 0.08 |
| TIES | 0.10 | 0.24 | 0.73 | 0 | 0.03 | 0.35 | 0.89 | 0 | 0.07 |
| DARE_TA | 0.36 | 0.85 | 0.98 | 0.12 | 0.29 | 0.91 | 1.00 | 0.21 | 0.33 |
| SCE | 0.01 | 0.06 | 0.54 | 0 | 0 | 0.22 | 0.49 | 0 | 0.01 |
| AF | 0.01 | 0.12 | 0.31 | 0 | 0 | 0.23 | 0.53 | 0 | 0.01 |
| **Merged models through CMM** | | | | | | | | | |
| CMM | **0.42** | 0.66 | 0.89 | 0.10 | 0.28 | 0.89 | 1.00 | 0.22 | 0.35 |
| DARE_CMM | 0.31 | 0.64 | 0.84 | 0.05 | 0.28 | 0.92 | 1.0 | 0.23 | 0.30 |

by effectively integrating their complementary strengths. Notably, CMM achieves the highest average accuracy of 61.43, surpassing KALE-LM-Chem-1.5 by 7.6% and the base model by 33.7%, and ranks first in 6 out of 9 tasks. DARE_CMM also achieves strong results, with an average accuracy of 59.93. These results demonstrate that CMM not only concentrates task-specific expertise but also mitigates inconsistencies, such as parameter interference and accumulated fine-tuning noise, that typically arise when combining heterogeneous Chemical LLMs. By progressively aligning expert models with the base model based on capability, CMM facilitates chemical knowledge generalization without sacrificing robustness. Compared to existing state-of-the-art model merging methods, including TA, TIES, DARE_TA, SCE, and AF, our approach delivers superior performance both in average and across individual tasks. While TA achieves a respectable average accuracies of 51.85, which is 18.5% lower than CMM, it also falls short on several tasks where our methods excel, such as Product_P, RS, and TP. Moreover, all baseline approaches struggle to resolve hidden noise and parameter conflicts, resulting in merged models that underperform even the best individual expert (KALE-LM-Chem-1.5). This indicates their limited effectiveness in preserving knowledge and managing inter-model inconsistencies. In contrast, CMM and DARE_CMM maintain consistently high performance across tasks, highlighting their effectiveness in preserving useful knowledge while avoiding performance degradation.

Table 2 reports the performance on the Mol-Instructions benchmark, which focuses on generative chemical tasks, specifically retrosynthesis and forward reaction prediction. In contrast to ChemBench, an **expert-abundant** benchmark with multiple experts demonstrating superior performance, Mol-Instructions presents an **expert-sparse** scenario, where only a single expert, Molinst-Molecule-8B, shows promising results. Specifically, it achieves round-trip accuracies of 0.42 and 0.31 on retrosynthesis and forward prediction, respectively, while other expert models, including those that performed well on ChemBench, exhibit near-zero performance on round-trip accuracy. This large performance disparity highlights the unique specialization of Molinst-Molecule-8B and the uneven distribution of expertise among the expert model group. Despite this imbalance, our proposed methods, CMM and DARE_CMM, successfully concen-

Table 3: The overall average performance across 9 prediction tasks from ChemBench and 2 generative tasks from Mol-Instructions.

| Models | Overall average |
|---|---|
| **Expert models** | |
| Llama-3-8B-Instruct | 38.85 |
| Molinst-molecule-8b | 41.79 |
| KALE-LM-Chem-1.5 | 47.97 |
| Meerkat-8b-v1.0 | 35.71 |
| GeLLMO-P6 | 27.53 |
| **SOTA merge methods** | |
| TA | 43.88 |
| TIES | 34.69 |
| DARE_TA | 30.86 |
| SCE | 32.84 |
| AF | 42.26 |
| **Merged models through CMM** | |
| CMM | 56.62 |
| DARE_CMM | 54.40 |

trate the rare but critical generative capability of Molinst-Molecule-8B without being negatively impacted by the poor performance of other experts. CMM achieves a round-trip accuracy of 0.42 on retrosynthesis, matching the top-performing expert, and maintains strong validity and SELFIES BLEU scores. On forward reaction prediction, CMM attains a round-trip accuracy of 0.28, closely approaching Molinst-Molecule-8B's 0.31. Importantly, further analysis reveals that CMM correctly generates a distinct subset of samples compared to Molinst-Molecule-8B, suggesting that the merged model does not merely inherit expert knowledge in a one-to-one fashion. While some knowledge from Molinst-Molecule-8B is not retained during the merging process, CMM also appears to generalize relevant patterns beyond individual model boundaries, bridging predictive reasoning and generative modeling. This finding underscores the advantage of CMM's capability-aware merging strategy, which not only preserves task-specialized knowledge but also supports meaningful high-level expert intelligence fusion. On the contrary, baseline model merging methods struggle in this expert-sparse setting. Their round-trip accuracy on retrosynthesis rarely exceeds 0.1, and drops to near zero on forward reaction prediction. This failure stems from their inability to handle highly imbalanced conditions where most experts contribute noise rather than signal, leading to poor knowledge integration and degraded performance.

To enable a unified comparison of model performance across the two benchmarks, we normalize the round-trip accuracy scores from Mol-Instructions by rescaling them to a 0–100 range. We then combine the results from the nine predictive tasks in ChemBench and the two generative tasks in Mol-Instructions to compute an overall average score, as shown in Table 3. CMM achieves the highest overall average score of 56.62, significantly outperforming all individual expert models and state-of-the-art model merging methods. Specifically, it exceeds the best-performing expert model, KALE-LM-Chem-1.5, by 18.03%, and outperforms the best baseline merging method, TA, by 29.03%. The DARE-CMM model also performs strongly, achieving an overall average score of 54.40, ranking second among all merged models and outperforming all expert models and other baseline approaches. These results clearly demonstrate the effectiveness of our progressive, capability-aware merging strategy for integrating chemical LLMs. Notably, CMM not only concentrates task-specific expertise but also mitigates inconsistencies and achieves strong generalization across both predictive and generative tasks—under both expert-abundant and expert-sparse conditions.

## 4.3 Analysis

We use the Subspace Alignment Ratio (SAR) [36] to investigate how the expert models are integrated into the merged model during the merging process. The SAR is used to quantify the alignment between the subspaces of two task matrices. The formal definition and computation details of SAR are provided in Appendix F. A higher SAR value indicates a greater overlap between the subspaces, meaning that the merged model better inherits the capabilities of the expert model.

We observe that as more expert models are merged later in the process, the SAR values of those merged earlier tend to slightly decrease. Additionally, higher assigned weights are generally associated with higher SAR values. This suggests that placing better-performing expert models later in the merging order, and assigning them higher weights, leads to higher SAR values. Take a model as an example. Table 4 shows the evolution of the SAR value for Molinst-molecule-8B during the CMM process. This expert model was merged at step 3, where its SAR value increased significantly. At step 4, the SAR value slightly decreased due to the merging of KALE-LM-Chem-1.5. When we reduced the weight assigned to Molinst-molecule-8B at step 3, we observed a corresponding decrease in its SAR value.

Table 4: SAR for Molinst-molecule-8B

| Metric | Step1 | Step2 | Step3 | Step4 | Step3(reduced weight) |
|--------|-------|-------|-------|-------|------------------------|
| SAR    | 0.042 | 0.042 | 0.650 | 0.647 | 0.506                  |

## 4.4 Ablation study

**Influence of the number of expert models** Starting from the full expert model set (GeLLMO-P6-Llama, Meerkat-8B-v1.0, Molinst-Molecule-8B, and KALE-LM-Chem-1.5), we progressively reduce the number of source models to evaluate how performance degrades across both predictive and

Table 5: Ablation study on the number of expert models used in CMM. We progressively reduce the number of merged expert models to evaluate the impact on CMM's performance across the ChemBench and Mol-Instructions benchmarks.

| GeLLMO | Meerkat | Molinst-molecule | KALE-LM-Chem | Chembench Average | Mol-Instructions Average | Overall Average |
|---|---|---|---|---|---|---|
| ✓ | ✓ | ✓ | ✓ | 61.43 | 0.35 | 56.62 |
| ✓ | ✓ | ✓ | | 48.47 | 0.13 | 42.02 |
| ✓ | ✓ | | | 48.88 | 0 | 40.00 |
| ✓ | | | | 49.34 | 0 | 40.37 |

Table 6: Ablation study on the expert model merging order. We compare the performance of CMM under different merging sequences, including reverse and random orders.

| Merge Sequence | Chembench Average | Mol-Instructions Average | Overall Average |
|---|---|---|---|
| Default | 61.43 | 0.35 | 56.62 |
| Reverse | 48.15 | 0.13 | 41.67 |
| Random | 52.05 | 0.26 | 47.31 |

generative benchmarks, results presented in Table 5. The full CMM configuration, which includes all four expert models, achieves the most superior performance on both benchmarks, with an average score of 61.43 on ChemBench and 0.35 on Mol-Instructions, resulting in the best overall average of 56.62. Removing the strongest predictive expert, KALE-LM-Chem-1.5, leads to a substantial drop in performance, ChemBench drops to 48.47 and Mol-Instructions to 0.13. Although KALE-LM-Chem-1.5 is a key contributor in boosting merging performance, the model after merging KALE-LM-Chem-1.5 achieves a siginificantly higher overall average score of 56.62 compared to KALE-LM-Chem's 47.97. This suggests that CMM enables a bi-directional improvement, where not only does the high-performing expert contribute significantly, but the inclusion of moderate-performing experts in the early stage also accumulate positive effect leading to the ultimate superior overall performance. As more experts are excluded, performance continues to decline, particularly in the generative domain where Mol-Instructions scores fall to zero once Molinst-Molecule is removed.

**Influence of the expert model merging order** To investigate the effect of merge sequence on model performance, we compared our default merge order in the main experiment with two alternative strategies: reverse order and random order. Table 6 presents an ablation study evaluating the impact of expert model merging order on the performance of CMM. The default merging sequence, determined based on model task-specific performance, achieves the best overall results, with an average score of 61.43 on ChemBench, 0.35 on Mol-Instructions, and an overall average of 56.62. In contrast, reversing the merging order results in a sharp decline in performance, particularly on ChemBench, dropping to 48.15, and overall, down to 41.67. Similarly, using a random merging order yields a moderate degradation in performance, with overall accuracy reduced to 47.31. To further understand these effects, we visualize and compare the cumulative performance trajectories under the default and reverse merging orders. As shown in Figure 2, the default order produces a steady and consistent performance gain, culminating in the highest final score after all expert models are merged. In contrast, the reverse order exhibits a diminishing return pattern: although merging the strongest expert model (KALE-LM-Chem) first provides an early boost, subsequent merging steps result in stagnant or reduced gains. This supports the hypothesis that progressively building the model's capacity using increasingly capable experts helps consolidate task-specific strengths while maintaining base model's adaptive capability for the next round.

**Influence of the merging strategy and weight coefficient** $\beta$ Since the expert models are ordered from weakest to strongest based on their individual performance, the distribution of $\beta$ is expected to be either uniform or gradually increasing. In this experiment, we evaluate three distribution strategies: constant, linear, and exponential. As shown in Table 7, the linear distribution yields the best overall performance. In the constant setting, all expert models are assigned the same $\beta_k$, which we set to 0.5—a value empirically close to the optimal according to prior research. Assigning equal weights to all expert models results in a notably lower overall average of 54.60. To further examine the effect of the coefficient range, we conducted an ablation study by varying $\beta$ within different intervals. In our experiments, we found that slightly increasing the coefficient could lead to marginal performance improvements, whereas slightly decreasing it often resulted in a noticeable drop in performance.

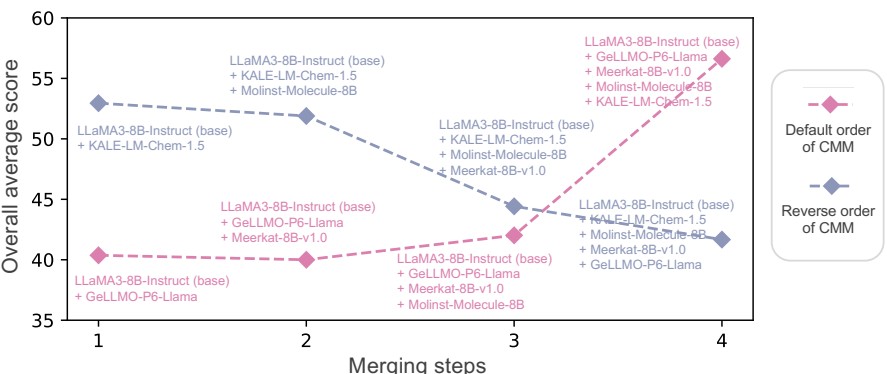

Figure 2: Performance comparison of the merged models along the trajectories of default/reverse order of CMM.

Table 7: Ablation study on the influence of the merging strategy and the coefficient range of weight $\beta$. We compare constant, linear, and exponential distributions of $\beta$, and further analyze the effect of varying the coefficient range on model performance.

| $\beta$ distribution | Chembench Average | Mol-Instructions Average | Overall Average |
|---|---|---|---|
| constant[0.3, 0.6] | 60.51 | 0.28 | 54.60 |
| linear[0.3, 0.6] | 61.43 | 0.35 | 56.62 |
| exponential[0.3, 0.6] | 61.38 | 0.33 | 56.22 |
| linear[0.2, 0.5] | 61.03 | 0.24 | 54.21 |
| linear[0.4, 0.7] | 60.38 | 0.41 | 56.86 |

## 4.5 Scalability

To evaluate the scalability and generality of CMM, we further apply it to settings beyond the chemical domain. The results demonstrate that CMM remains effective and stable under these extended settings, indicating that the underlying merging mechanism is not limited to chemistry-specific tasks. Detailed experimental setups and results are provided in Appendix E.

## 5 Conclusion

In this work, we propose Curriculum Model Merging (CMM), a progressive, capability-aware framework tailored to the challenges of merging heterogeneous chemical LLMs. CMM addresses the domain-specific issues of model disparity and imbalanced task coverage by structuring the merging process as a curriculum and applying adaptive weighting to expert contributions. Evaluations on ChemBench and Mol-Instructions show that CMM consistently outperforms individual experts and state-of-the-art baselines, while generalizing well across predictive and generative tasks in both expert-abundant and expert-sparse settings. These results demonstrate CMM's effectiveness as a scalable and privacy-conscious approach for building versatile chemical language models from specialized experts.

## 6 Limitation

While our approach demonstrates strong empirical performance, several limitations remain. First, model merging currently requires all expert models to share the same architecture, which may limit its applicability to settings with different model architectures. Second, performance can also degrade when merging a large number of expert models. This is a known challenge observed in prior work [50]. Lastly, although initial analyses suggest why CMM achieves improved results, a deeper theoretical understanding remains future research.

## Acknowledgements

We thank all the anonymous reviewers for their constructive suggestions on improving this paper. This paper is partially supported by National Natural Science Foundation of China (NSFC) grant 62441236.

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

# A Large language models (LLMs) in the chemical domain

Table 8 presents a brief overview of recently published chemical LLMs.

Table 8: Summary of LLMs in the chemical domain

| Model | Time | #Parameters | Base model | Dataset | Capability |
|---|---|---|---|---|---|
| MolXPT [33] | 2023.05 | 350M | GPT-2 | PubChem, PubMed | Mol. und. |
| CrystaLLM [1] | 2023.07 | - | GPT-2 | MP, OQMD, NOMAD | Crystal gen. |
| DARWIN-MDP [53] | 2023.08 | 7B | LLaMA | SciQ, FAIR | Sci. |
| Mol-Instructions [15] | 2023.11 | 7B/8B | LLaMA-2/LLaMA-3 | Mol-Instructions | Mol. gen. |
| ChemDFM [63] | 2024.01 | 8B/13B | LLaMA-3/LLaMa | Chemical literature, textbooks | Mol. und. |
| SciGLM [61] | 2024.01 | 6B | ChatGLM3 | SciInstruct | Sci. |
| ChemLLM [62] | 2024.02 | 7B | InternLM-2 | ChemData and Multi-Corpus | Prop. pred. |
| LlaSMol [59] | 2024.02 | 7B | Mistral | SMolInstruct | Prop. pred. |
| ProLLaMA [35] | 2024.02 | - | LLaMA-2 | UniRef50 | Prot. und. |
| ProtLLM [66] | 2024.02 | 7B | LLaMA | InterPT | Prot. gen. |
| Meerkat [28] | 2024.04 | 7B/8B/70B | Mistral/LLaMA-3/LLaMA-3 | 18 medical textbooks | Med. |
| DrugLLM [31] | 2024.05 | 7B | LLaMA | ZINC, ChEMBL | Mol. und. |
| MolecularGPT [32] | 2024.06 | 7B | LLaMA-2 | QM9, ChEMBL | Prop. pred. |
| SciLitLLM [30] | 2024.08 | 7B/14B | Qwen2 | SciLitIns | Sci. |
| KALE-LM [9] | 2024.09 | 8B | Llama-3.1 | - | Prop. pred. |
| LLAMAT [37] | 2024.12 | 7B/8B | LLaMA-2/LLaMA-3 | R2CID | Mater. pred. |
| GeLLMO [11] | 2025.02 | 7B/8B | Mistral-v0.3/Llama-3.1 | MuMOInstruct | Mol. gen. |
| Mol-LLaMA [27] | 2025.02 | 8B | Llama-3.1 | Mol-LLaMA-Instruct | Mol. und. |

# B Comparison between CMM and other machine learning methods

To demonstrate the advantage of CMM, we provide the results of comparisons between CMM and other machine learning methods. We conducted an additional experiment where we supervised fine-tuned (SFT) Llama-3-8B-Instruct on the dev set of ChemBench and compared it with our CMM model. Due to the limited training data, the SFT model performed poorly, as shown in Table 9. We also compare CMM with GPT-4 [40], a leading closed-source model, based on results reported in prior work [62]. This comparison reveals that our CMM model achieves competitive performance with GPT-4. While GPT-4 remains stronger overall performance(65.89 vs. 61.43), CMM offers a compelling trade-off by delivering strong performance with zero training cost, full reproducibility, and no reliance on proprietary APIs or data.

Table 9: Comparison between CMM and other machine learning methods

| Model | Chembench Avg. |
|---|---|
| SFT | 19.19 |
| GPT-4 | 65.89 |
| CMM | 61.43 |

# C Influence of different score computation methods

In the main experiment, the overall performance score for each model is computed by first normalizing its scores across all tasks and then taking the average. Here we propose an alternative method for computing the overall performance scores. For each model, we first compute the average score within each benchmark, where each benchmark consists of multiple tasks. These per-benchmark averages are then normalized to ensure comparability, and their mean is taken as the overall performance score. Under this alternative computation scheme, the ranking of expert models from weakest to strongest is as follows: GeLLMO-P6-Llama, Meerkat-8B-v1.0, KALE-LM-Chem-1.5, and Molinst-Molecule-8B. Keeping other settings unchanged, we apply the Curriculum Model Merging (CMM) procedure based on this order. For comparison, we also recompute the overall scores of the CMM obtained in the main experiment using the new scoring method. The results are summarized in Table 10. This method of computing overall scores overlooks the variation in the number of tasks across different benchmarks and therefore cannot accurately reflect the capabilities of each model. As a result, the performance of

the merged model obtained in this setting is inferior to that of the original one. Specifically, while the average score on Mol-Instructions remains similar, there is a notable drop in performance on ChemBench.

Table 10: Ablation study on different score computation methods

| Model | Chembench Average | Mol-Instructions Average | Overall Average |
|---|---|---|---|
| **Expert models** | | | |
| Llama-3-8B-Instruct | 45.93 | 0.07 | 26.47 |
| Molinst-molecule-8b | 42.97 | 0.37 | 39.99 |
| KALE-LM-Chem-1.5 | 57.07 | 0.07 | 32.04 |
| Meerkat-8b-v1.0 | 43.65 | 0 | 21.83 |
| GeLLMO-P6 | 31.87 | 0.08 | 19.94 |
| **Merged model** | | | |
| CMM | 61.43 | 0.35 | 48.22 |
| CMM_AltRank | 53.02 | 0.38 | 45.51 |

# D    Relationship between the number of samples and results

Figure 3 provides more details of the impact of the number of evaluation samples on the resulting evaluation metrics. The evaluation metrics do not exhibit significant upward or downward trends beyond 200 samples. Therefore, we randomly selected 200 evaluation samples to improve computational efficiency.

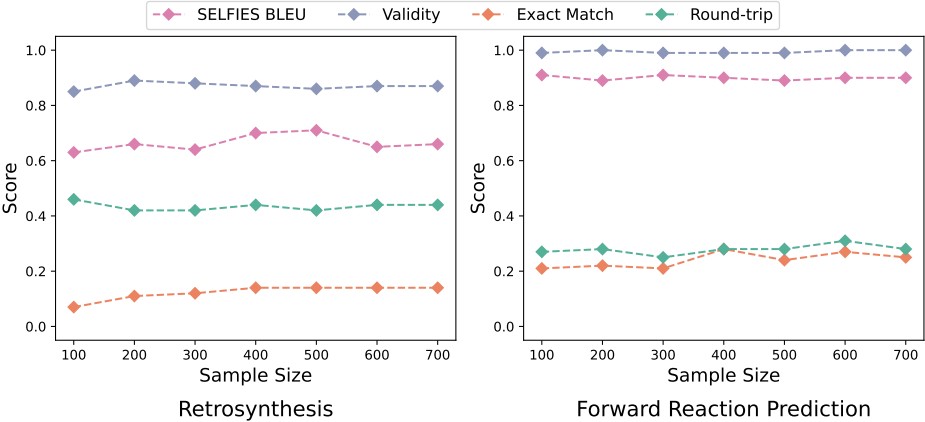

Figure 3: Relationship between the number of evaluation samples and the resulting metrics

# E    Scalability of CMM

## E.1    General model set

CMM is generalizable and applicable beyond the chemical domain. We also applied CMM to general-purpose LLMs and still achieved the best performance. In this experiment, we adopted the universal Llama-3.1-8B-Instruct [18] as the base model, and Hermes-3-Llama-3.1-8B [45], Llama-3.1-Tulu-3-8B [29], Llama-3.1-Storm-8B [3], and Llama-3.1-SuperNova-Lite [2] as expert models, following this merging order. The performance of the models was evaluated across 7 benchmarks spanning multiple domains, including general, instruction following, mathematics, and coding: MMLU [20], IFEval [65], ARC-C [56], DROP [13], BBH [41], GSM8K [6], and Math [21]. For comparison, we also included the model generated using the Task Arithmetic (TA) method.

The model merged using CMM outperforms all expert models and the task arithmetic baseline in terms of average score across the seven benchmarks. Detailed results are presented in Table 11.

Table 11: CMM on a general model set

| Model | MMLU | IFEval | ARC-C | DROP | BBH | GSM8K | Math | Avg |
|---|---|---|---|---|---|---|---|---|
| **Expert models** | | | | | | | | |
| Llama-3.1-8B-Instruct | 69.35 | 68.11 | 81.69 | 82.33 | 57.61 | 84.15 | 39.76 | 69.00 |
| Hermes-3-Llama-3.1-8B | 64.29 | 59.59 | 82.03 | 77.73 | 62.25 | 81.50 | 25.20 | 64.66 |
| Llama-3.1-Tulu-3-8B | 66.26 | 75.18 | 66.10 | 76.68 | 61.18 | 87.87 | 40.02 | 67.61 |
| Llama-3.1-Storm-8B | 68.87 | 72.18 | 81.69 | 79.01 | 56.01 | 82.94 | 35.60 | 68.04 |
| Llama-3.1-SuperNova-Lite | 69.29 | 69.90 | 84.07 | 81.84 | 60.10 | 83.40 | 36.58 | 69.31 |
| **Merge methods** | | | | | | | | |
| TA | 67.96 | 67.03 | 82.03 | 80.02 | 69.80 | 81.43 | 41.10 | 69.91 |
| CMM | 69.81 | 70.26 | 81.69 | 83.07 | 61.11 | 84.61 | 40.96 | 70.22 |

## E.2 Larger and more diverse model set

To evaluate the scalability of CMM, we extended our original experiment by introducing two additional tasks—reasoning and math—evaluated on DROP [13] and GSM8K [6], respectively. We also included two high-performing expert models on these tasks: Llama-3.1-SuperNova-Lite [2] and Llama-3.1-Tulu-3-8B [29]. For comparison, we also included the model generated using the Task Arithmetic (TA) method.

The performance of the six expert models and the CMM-merged model across 13 tasks is shown in Table 12. The CMM-merged model outperforms all expert models as well as the TA-merged model in terms of overall performance. This demonstrates the effectiveness of CMM when applied to larger and more diverse model sets.

When examining individual benchmarks, we observe that the CMM merged model not only inherits but also surpasses the best expert performance on Chembench, indicating strong domain adaptation capabilities. However, on the other three benchmarks—Mol-Instructions, DROP, and GSM8K—CMM does not outperform the top-performing expert models, suggesting that while CMM effectively aggregates knowledge, certain task-specific expertise may be partially diluted during the merging process. We would like to clarify that this phenomenon is not specific to CMM, but rather a common limitation across model merging methods. Prior research has shown that even state-of-the-art merging methods often experience performance saturation after merging no more than six expert models [50]. This highlights an open challenge in the field and calls for further theoretical understanding and methodological advances to improve the scalability of model merging.

Table 12: CMM on a larger and more diverse model set

| Model | Chembench Avg | Mol-Instructions Avg | DROP | GSM8K | Overall Avg |
|---|---|---|---|---|---|
| **Expert models** | | | | | |
| Llama-3-8B-Instruct | 45.93 | 0.07 | 18.63 | 79.38 | 40.41 |
| GeLLMO-P6 | 31.87 | 0.08 | 81.91 | 84.53 | 36.10 |
| Meerkat-8b-v1.0 | 43.65 | 0 | 74.3 | 44.81 | 39.38 |
| Molinst-molecule-8b | 42.97 | 0.37 | 24.15 | 62.62 | 42.12 |
| SuperNova-8B | 52.5 | 0 | 81.84 | 83.40 | 49.06 |
| Llama-Tulu-3-8B | 53.28 | 0 | 76.68 | 87.87 | 49.54 |
| KALE-LM-Chem-1.5 | 57.07 | 0.07 | 56.72 | 67.63 | 50.15 |
| **Merge methods** | | | | | |
| TA | 27.35 | 0.15 | 62.50 | 66.19 | 31.14 |
| CMM | 63.80 | 0.19 | 67.84 | 79.15 | 58.40 |

# F  Subspace Alignment Ratio

Subspace Alignment Ratio(SAR) [36] is defined as:

$$\text{SAR}(\Delta_t, \Delta_M; k_M) = \frac{\|\Pi_{k_M, M}\Delta_t\|_F}{\|\Delta_t\|_F}, \tag{9}$$

where $\Delta_t$ denotes a task vector, $\Delta_M$ is the difference between the parameters of the merged model and those of the base model, and $\Pi_{k_M, M}$ is the projection matrix onto the subspace spanned by the top $k_M$ left-singular vectors of $\Delta_M$. The number of singular vectors used ($k_M$) is formulated as:

$$k_M = \min\left\{k : \frac{\sum_{i=k+1}^{r}\sigma_i^2}{\sum_{i=1}^{r}\sigma_i^2} \leq \epsilon^2\right\} \tag{10}$$

where $\sigma_i$ denotes the singular values of $\Delta_M$, and $\epsilon = 0.05$.

