# OpenReview forum: "Curriculum Model Merging: Harmonizing Chemical LLMs for Enhanced Cross-Task Generalization"
_NeurIPS.cc/2025/Conference — NeurIPS 2025 poster_

### Official Review · Reviewer_Chme · 2025-06-21

**Clarity:** 2
**Significance:** 2
**Originality:** 2
**Rating:** 5
**Confidence:** 3

**Summary:**

In this paper, the authors introduce a novel model merging approach known as Curriculum Model Merging (CMM), which progressively integrates expert chemical Large Language Models (LLMs) in a gradual and continuous manner. This method comprises two primary steps: First, the models are ranked according to their capabilities, and a merge order is established. Second, an iterative merging procedure is executed. The experimental results on ChemBench and Mol-instructions demonstrate the effectiveness of the proposed approach.

**Questions:**

1. The notation $\lambada$ and $\tau_{new}$ in Sec.3.1 should be clearly described.
2. According to Tab. 1, the capabilities of Meerkat and GeLLMO are both worse than the base model LLaMA-3. Why were they selected for merging in the experiments? Would the merged model perform better if they were excluded?
3. It can be observed in Tab. 4 that the performance on ChemBench improves after removing MolInst-Molecule and Meerkat. More discussion on this observation would be beneficial.
4. There are some typos. For instance, "Tab. 4.2" in lines 170 and 284 should be corrected to "Tab. 3." The authors are advised to thoroughly review the paper for consistency and accuracy.

**Ethical Concerns:**

["NO or VERY MINOR ethics concerns only"]

**Final Justification:**

After considering the rebuttal and discussions, I find that the experimental results address my concerns to a large extent. I have carefully read the authors' response as well as the discussion with other reviewers. The rebuttal meaningfully improves the paper’s clarity and technical soundness. I have decided to raise the score.

**Limitations:**

The authors claim to have discussed the limitations in Section 5. However, the reviewer did not notice any such discussion in the conclusion section. It would be beneficial to include a clear discussion of the limitations in the appropriate section, such as the conclusion or a dedicated limitations section.

**Quality:**

3

**Strengths And Weaknesses:**

The paper is well-organized, and the proposed approach is technically sound. Model merging is an interesting yet challenging task that enables the integration of multiple expert LLMs into a unified model without requiring access to the original training data or incurring additional computational costs. The proposed approach is simple but effective for practical applications.

However, the reviewer still has some concerns. The notation in Sec. 3.1 should be clearly described, for example, $\lambda$ and $\tau_{new}$. It would be beneficial to provide more in-depth discussion on the experimental results. Additionally, the proposed approach has only been evaluated on LLaMA-3 based models for chemical tasks. The paper should be carefully reviewed to avoid minor errors.

---

> ### Author Rebuttal · Authors · 2025-07-31
>
> We appreciate very much your constructive comments on our paper. Please kindly find our response to your comments below. We hope that our response satisfactorily addresses the issues you raised. Please feel free to let us know if you have any additional concerns or questions.
>
> **Q1**: The notation $\lambda$ and $\tau_{new}$ in Sec.3.1 should be clearly described.
> >Thank you for pointing this out.
> > - $\lambda$ is an optional scaling term, it is determined using validation sets.
> > - $\tau_{new}$ is the sum of all task vectors.
>
> **Q2**: The capabilities of Meerkat and GeLLMO are both worse than the base model LLaMA-3. Why were they selected for merging in the experiments? Would the merged model perform better if they were excluded?
> >Thank you for the insightful question.
> > - The merged model **wouldn't perform better** if they were excluded. The result is following:
> >   | Model   | Chembench Average | Mol-Instructions Average |  Overall Average |
> >   | -------- | ------ | ------ |  ----- |
> >   | Merged model(without GeLLMO and Meerkat)   | 61.41   | 0.35   |  56.52 |
> > - Excluding Meerkat and GeLLMO results in an overall score of **56.62**, which is slightly lower than that of the original merged model. Although Meerkat and GeLLMO perform relatively poorly on the selected benchmarks, they carry complementary chemical knowledge. This additional knowledge contributes to the construction of chemical understanding in the merged model.
> > - Similarly, previous research has demonstrated that **"strong student models can benefit from learning even from weaker teacher models"**[1].
> >
> >[1] Hunter et al. "Theoretical Analysis of Weak-to-Strong Generalization."
>
> **Q3**: The performance on ChemBench improves after removing MolInst-Molecule and Meerkat. More discussion on this observation would be beneficial.
> >Thanks for your valuable comment regarding the issue of conflicts in model merging.
> > - In model merging, it is often difficult to avoid interference with previously acquired capabilities when new ones are introduced. This is because **key parameters related to earlier capabilities may be modified during the merging process**. In our experiments, the CMM-merged model successfully acquired the ability to solve Mol-Instructions tasks, while the performance on ChemBench experienced only a slight drop. We consider this trade-off acceptable, as **it reflects a balance aimed at optimizing the model's overall performance**.
>
> **Q4**: There are some typos. For instance, "Tab. 4.2" in lines 170 and 284 should be corrected to "Tab. 3."
> > Thank you for pointing this out. We have carefully reviewed the paper and corrected all these typos.
>
> **W1**: The proposed approach has only been evaluated on LLaMA-3 based models for chemical tasks.
> > We appreciate the reviewer’s thoughtful comment.
> > - CMM is **generalizable** and **applicable** beyond the chemical domain. We also applied CMM to **general**-purpose LLMs and still achieved the best performance, as outlined below.
> >   - We adopted the universal **Meta-Llama-3.1-8B-Instruct**[1] as the base model, and used **Hermes-3-Llama-3.1-8B**[2], **Llama-3.1-Tulu-3-8B**[3], **Llama-3.1-Storm-8B**[4], and **Llama-3.1-SuperNova-Lite**[5] as expert models.
> >   - The performance of the models was evaluated across 7 benchmarks spanning multiple domains, including **general**, **instruction following**, **mathematics**, and **coding**. For comparison, we also included models generated using the Task Arithmetic (TA) method.
> >   - The model merged using CMM **outperforms** all expert models and the task arithmetic baseline in terms of average score across the seven benchmarks.
> >   | Model                             | MMLU  | IFEval | ARC-C | DROP  | BBH   | GSM8K | Math  | Avg   |
> >|----------------------------------|-------|--------|-------|-------|-------|-------|--------|--------|
> >| Meta-Llama-3.1-8B-Instruct       | 69.35 | 68.11  | 81.69 | 82.33 | 57.61 | 84.15 | 39.76 | 69.00 |
> >| Hermes-3-Llama-3.1-8B            | 64.29 | 59.59  | 82.03 | 77.73 | 62.25 | 81.50 | 25.20 | 64.66 |
> >| Llama-3.1-Tulu-3-8B_hf           | 66.26 | 75.18  | 66.10 | 76.68 | 61.18 | 87.87 | 40.02 | 67.61 |
> >| Llama-3.1-Storm-8B               | 68.87 | 72.18  | 81.69 | 79.01 | 56.01 | 82.94 | 35.60 | 68.04 |
> >| Llama-3.1-SuperNova-Lite         | 69.29 | 69.90  | 84.07 | 81.84 | 60.10 | 83.40 | 36.58 | 69.31 |
> >| TA                               | 67.96 | 67.03  | 82.03 | 80.02 | 69.80 | 81.43 | 41.10 | 69.91 |
> >| CMM                              | 69.81 | 70.26  | 81.69 | 83.07 | 61.11 | 84.61 | 40.96 | **70.22** |
> >
> >[1] Team Llama et al. " The llama 3 herd of models."
> >
> >[2] Teknium et al. "Hermes 3 Technical Report."
> >
> >[3] Nathan et al. "Tulu 3: Pushing Frontiers in Open Language Model Post-Training."
> >
> >[4] Ashvini Kumar Jindal. "Llama-3.1-Storm-8B: Improved SLM with Self-Curation + Model Merging."
> >
> >[5] Lucas et al. "Arcee-SuperNova: Training Pipeline and Model Composition."
>
> **Limitations**: It would be beneficial to include a clear discussion of the limitations in the appropriate section, such as the conclusion or a dedicated limitations section.
> >Thank you for pointing this out. We have now included a dedicated "Limitations" section in our revised paper. The main limitations of our current work are as follows:
> > - **Homogeneous architecture constraint**: Model merging requires all expert models to share the same architecture, which limits its applicability to broader scenarios involving heterogeneous model types.
> > - **Scalability with many expert models**: When merging a large number of expert models, performance tends to degrade. As discussed in prior work, this is a **known limitation** of model merging in general, not specific to our method.
> > - **Theoretical understanding and extensibility**: While there has been some exploration into why CMM outperforms existing merging methods, the theoretical foundations are still not fully understood. Further research is needed to deepen our understanding and to develop methods that can integrate a larger number of expert models while preserving high performance.

---

> > ### Comment · Reviewer_Chme · 2025-08-02
> >
> > Thank you for your reply. The experimental results address my concerns to a large extent.

---

> > > ### Author Response · Authors · 2025-08-06
> > >
> > > Thank you for your prompt response and for acknowledging the addressed concerns. We sincerely appreciate your time and feedback on our work.

---

### Official Review · Reviewer_1evb · 2025-07-01

**Clarity:** 3
**Significance:** 3
**Originality:** 3
**Rating:** 4
**Confidence:** 4

**Summary:**

This paper presents Curriculum Model Merging (CMM), a novel framework for combining multiple, specialized Large Language Models (LLMs) in the chemical domain into a single, more versatile model. The authors identify that directly merging chemical LLMs is challenging due to significant disparities among models fine-tuned for specific tasks and an imbalanced distribution of expertise across the field.
The core idea of CMM is to merge models progressively, following a curriculum. The key contributions of the paper are --

A Novel Merging Framework: CMM first evaluates and ranks the individual expert models based on their performance. It then iteratively merges them, starting from the weakest-performing model and gradually incorporating stronger ones. This step-by-step process aims to harmonize the models, reduce parameter interference, and preserve specialized knowledge.

State-of-the-Art Performance: The paper demonstrates through extensive experiments that CMM significantly outperforms existing model merging methods. On a combined set of predictive and generative tasks, CMM achieves an overall score 29.03% higher than the next best baseline method.

Robust Generalization: The proposed method proves effective in two distinct scenarios: an "expert-abundant" setting with many models available and an "expert-sparse" setting where only one model possesses the critical expertise for a task. CMM successfully consolidates knowledge for both predictive and generative tasks without significant performance degradation.

Practical Application: This work is the first to systematically adapt model merging for the chemical domain, offering a practical, privacy-preserving solution that does not require access to original training data or incur substantial computational costs

**Questions:**

1. On the Generalizability of CMM to Architecturally Diverse Models --
Your experiments compellingly demonstrate CMM's effectiveness when merging models from the LLaMA family (LLaMA-2, LLaMA-3, Mistral). This architectural homogeneity raises a question about the broader applicability of the CMM framework. Could you discuss the anticipated challenges or necessary modifications for applying CMM to a more architecturally diverse set of models? For instance, what would be the implications of merging a traditional transformer-based LLM with a model based on a graph neural network, another common architecture in the chemical domain?

2. Providing Concrete Evidence for Synergistic Knowledge Generalization --
The paper makes the exciting claim that the CMM-merged model "does not merely inherit expert knowledge" but "appears to generalize relevant patterns beyond individual model boundaries". This suggests a synergistic effect where the whole is greater than the sum of its parts. To make this powerful claim more concrete, could you provide a few qualitative examples from your test set?

3. Regarding the Robustness of the Single-Metric Ranking Curriculum --
The curriculum in CMM is constructed based on a single, overall performance score that is aggregated across multiple tasks. While your ablation study explores different score computation methods, it still relies on a single final metric for ranking. Have you considered scenarios where this single metric might be insufficient? For example, a model could have a low overall score but be the only expert on a rare but critical task. How would CMM ensure this vital, niche capability is not diluted or lost when it is merged early in the curriculum as a "weaker" model?

4. Elucidating the Choice of Merging Weight Hyperparameters --
Your ablation study on the merging weight distribution (constant, linear, exponential) is very informative. The paper states that the best-performing linear strategy used coefficients that increased from 0.3 to 0.6. Could you provide more intuition on how sensitive the model's performance is to this specific range? How was this range determined?

**Ethical Concerns:**

["NO or VERY MINOR ethics concerns only"]

**Limitations:**

1. Discussion of Methodological and Experimental Limitations --
A dedicated limitations section should be added to address the following points:
Architectural Homogeneity: The current study exclusively uses expert models from the LLaMA family. The authors should acknowledge that the effectiveness of CMM on architecturally diverse models (e.g., merging a transformer with a graph neural network) has not been tested and may present unforeseen challenges.

Scalability of Sequential Merging: The experiments involve merging four expert models. The paper should discuss the potential scalability of the sequential CMM approach. As the number of expert models grows into the dozens or hundreds, the linear, one-by-one merging process could become a computational bottleneck, and potential "catastrophic forgetting" of the earliest merged models might become more pronounced.

Robustness of the Curriculum Ranking: The curriculum is based on a single aggregated performance score. The authors should discuss the potential brittleness of this approach. For example, a model that is weak overall but possesses a unique, critical capability might be merged too early, risking the dilution of its rare knowledge by subsequent, stronger models.

Hyperparameter Sensitivity: The paper shows a linear weighting strategy works best but could benefit from a discussion on the sensitivity and tuning process for the weight coefficients (e.g., the 0.3 to 0.6 range). This would provide a more complete picture of the practical implementation of CMM.

2. Discussion of Potential Negative Societal Impact

The development of more powerful and versatile chemical language models carries potential risks that warrant discussion:

Dual-Use Applications: A model that effectively consolidates knowledge from disparate chemical domains could be a powerful tool for beneficial research like drug discovery. However, it could also be misused to accelerate the design of harmful substances, such as novel toxins, illicit drugs, or chemical weapons. The authors should acknowledge this dual-use potential as a significant societal risk associated with creating more powerful, generalist chemical AI.

Over-Reliance and De-skilling: The availability of a highly capable, "all-in-one" chemical LLM could lead to over-reliance by researchers and practitioners. This could potentially de-skill human chemists or stifle creative, out-of-the-box human problem-solving. It also raises the risk of significant errors if the model's failure modes are not well-understood by its users.

Accessibility and Malicious Use: The CMM method is designed to be practical and efficient. This accessibility means that malicious actors could potentially use it to merge their own privately-trained expert models to create a powerful tool for nefarious purposes, bypassing the safeguards of public models. A brief discussion on the importance of responsible model governance would be appropriate.

**Quality:**

3

**Strengths And Weaknesses:**

Strengths -----
Quality: The paper demonstrates high quality through its rigorous and comprehensive experimental validation. The authors use two distinct benchmarks, Chembench and Mol-Instructions, which together cover 11 different chemical tasks, spanning both prediction and generation. The inclusion of extensive ablation studies systematically validates the core components of the CMM framework; for example, the performance drops significantly from 56.62 to 41.67 when the merging order is reversed, confirming the curriculum's importance. Furthermore, the comparison against multiple relevant and high-performing baseline methods like TA, TIES, and DARE_TA provides strong evidence for CMM's superior performance.

Clarity: The paper is clear and well-written. The core concepts are introduced logically, and the methodology is presented with clear mathematical formalism. A major strength is the use of visualizations to aid understanding. Figure 1 provides an excellent intuitive illustration of the CMM process compared to standard methods , while Figure 2 effectively visualizes the performance gains at each step of the merging curriculum, making a compelling case for the chosen merging order.

Significance: The work addresses a problem of high practical significance: creating a single, versatile chemical LLM from multiple specialized models without needing the original data or incurring massive computational costs. The demonstrated success of CMM in both "expert-abundant" (Chembench) and "expert-sparse" (Mol-Instructions) scenarios underscores its robustness and broad utility. The finding that CMM can preserve and even generalize rare knowledge—as seen in its performance on retrosynthesis where only one expert model was competent—is a particularly significant result. The overall performance improvement of 29.03% over the best SOTA baseline is substantial and impactful.

Originality: The paper's primary originality lies in being the first to systematically adapt and tailor model merging techniques to the unique challenges of the chemical LLM domain. The authors originally identify the key problems of high model disparity and imbalanced task coverage in chemistry. The proposed solution, framing the merging process as a curriculum where models are progressively integrated based on their capability, is a novel and insightful contribution that directly addresses these challenges.


Weaknesses -----
Quality: A potential critique of the paper's quality could center on the homogeneity of the models used. All the expert models chosen are based on the LLaMA architecture (LLaMA-2, LLaMA-3, Mistral). An argument could be made that the method's effectiveness has not been demonstrated on a more diverse set of model architectures, leaving its generalizability beyond LLaMA-like models in question. Additionally, the entire curriculum is constructed based on a single aggregated performance score, which could be a point of weakness if a model performs poorly on average but possesses unique, valuable capabilities on a niche task not well-represented in the overall score.

Clarity: While the paper's results are clear, the analysis of why the merged model generalizes could be deeper. The paper makes the exciting claim that the CMM-merged model "does not merely inherit expert knowledge" and "appears to generalize relevant patterns". However, this is not substantiated with qualitative examples or deeper analysis, leaving the inner workings of the knowledge fusion somewhat opaque.

Significance:  As all experiments and validation are confined to the chemical domain, the paper does not provide evidence that the CMM approach would be beneficial for merging models in other domains like computer vision or robotics, where the nature of expert specialization might differ.

Originality: An argument could be made that the paper's originality is somewhat limited, as it combines two existing lines of research: task-vector-based model merging (like Task Arithmetic) and curriculum learning. From this perspective, the core contribution could be framed as a clever and effective application of known concepts to a new domain, rather than the invention of a fundamentally new merging paradigm.

---

> ### Author Rebuttal · Authors · 2025-07-31
>
> Thank you sincerely for your thoughtful and positive feedback on our work. We are particularly grateful for your recognition of the various aspects of our research. Below, we have provided a detailed explanation for your remaining concern as follows. Please do not hesitate to let us know if you have any further questions.
>
> **Q1**:Could you discuss the anticipated challenges or necessary modifications for applying CMM to a more architecturally diverse set of models?
> > We appreciate your inquiry about merging architecturally diverse set of models.
> > - CMM belongs to the category of model merging, which can only integrate models with the same architecture. However, model fusion methods could be explored to combine an architecturally diverse set of models.
> > - Approaches for integrating multiple model capabilities into a single model can be categorized into two types.
> >   - The first is **model fusion**, which integrates **heterogeneous** models in a manner similar to knowledge distillation, requiring both training and data during the process.
> >   - The second is **model merging**, which combines only **homogeneous** models and does not require any training or data, as it simply involves combining model parameters.
>
> **Q2**: The paper makes the exciting claim that the CMM-merged model "does not merely inherit expert knowledge" but "appears to generalize relevant patterns beyond individual model boundaries". To make this powerful claim more concrete, could you provide a few qualitative examples from your test set?
> > We sincerely thank the reviewer for this valuable suggestion. In the benchmark, there are cases where none of the expert models answered a question correctly, but the CMM-merged model produced the correct answer. Below is one such example.
> > - Question: What substances are commonly selected for the synthesis of NC(=O)c1ccc(N2CCN(Cc3cn4cc(Cl)ccc4n3)CC2)c(Cl)c1 ?
> >A. There's a likelihood of reactions happening, with COc1ccc(-c2cc(C)cc(NC(=O)C3(c4ccc5c(c4)OC(F)(F)O5)CC3)n2)c(C)n1.Cl>C1COCCO1>NC(=O)c1ccc(N2CCN(Cc3cn4cc(Cl)ccc4n3)CC2)c(Cl)c1.
> >B. Undergoing reactions is a possibility, and N#Cc1ccc(N2CCN(Cc3cn4cc(Cl)ccc4n3)CC2)c(Cl)c1.[OH-].[Na+].CC(C)(C)O.CO>OO.ClCCl>NC(=O)c1ccc(N2CCN(Cc3cn4cc(Cl)ccc4n3)CC2)c(Cl)c1.
> >C. Components used in the production of COc1cccc2c1nc(C(F)F)n2-c1nc(N2CCOCC2)nc(N(CCCN(C)C)C2CCN(S(=O)(=O)CCl)CC2)n1.
> >D. The potential for reactions to materialize exists, with C/C(=N\O)c1ccc2c(c1)B(O)OC2(C)C>CC(=O)O.[Zn]>NC(=O)c1ccc(N2CCN(Cc3cn4cc(Cl)ccc4n3)CC2)c(Cl)c1.
>
> >   - The correct answer is B, while the expert models predicted C, C, A, and A, respectively. In contrast, the CMM-merged model selected the correct answer B, demonstrating that our merged model not only inherits knowledge from the experts, but also makes more accurate and informed decisions.
>
> **Q3**: A single final metric for ranking might be insufficient? For example, a model could have a low overall score but be the only expert on a rare but critical task. How would CMM ensure this vital, niche capability is not diluted or lost when it is merged early in the curriculum as a "weaker" model?
> > Thank you for your question. We agree that rare but critical capabilities should be preserved during the merging process, even if they come from expert models with relatively low overall scores.
> > - To address this question, we plan to adjust the ranking strategy by assigning **higher priority** to such expert models in order to prevent their capabilities from being merged too early and potentially diluted.
> > - Whether an early-merged rare capability is preserved also depends on its **orthogonality to the capabilities introduced by later models**. If the rare capability is largely independent of those introduced later, it tends to be less affected and better preserved in the merged model.
>
>
> **Q4**: The paper states that the best-performing linear strategy used coefficients that increased from 0.3 to 0.6. Could you provide more intuition on how sensitive the model's performance is to this specific range? How was this range determined?
> >We appreciate your inquiry about the impact of the merging coefficient $\beta$, and will include the following sensitivity analysis in the updated submission.
> > - We draw inspiration from a conclusion in task arithmetic: “Scaling coefficients in the range 0.3 to 0.5 produce close to optimal results in many cases” [1].
> > - In our experiments, we found that **slightly increasing** the coefficient could lead to **marginal performance improvements**, whereas **slightly decreasing** it often resulted in a **noticeable drop** in performance. Based on these observations, we decided to select coefficients from the range [0.3, 0.6], which is close to the originally suggested [0.3, 0.5] interval.
> >   | Range   | Chembench Average | Mol-Instructions Average |  Overall Average |
> >   | -------- | ------ | ------ |  ----- |
> >   | [0.4,0.7]   | 60.38   | 0.41   |  56.86 |
> >   | [0.3,0.6]   | 61.43   | 0.35   |  56.62 |
> >   | [0.2,0.5]   | 61.03   | 0.24   |  54.21 |
> >
> >[1]Ilharco et al. "Editing Models with Task Arithmetic."
>
>
> **Significance**: As all experiments and validation are confined to the chemical domain, the paper does not provide evidence that the CMM approach would be beneficial for merging models in other domains.
> > We appreciate the reviewer’s thoughtful comment.
> > - CMM is **generalizable** and **applicable** beyond the chemical domain. We also applied CMM to **general**-purpose LLMs and still achieved the best performance, as outlined below.
> >   - We adopted the universal **Meta-Llama-3.1-8B-Instruct**[1] as the base model, and used **Hermes-3-Llama-3.1-8B**[2], **Llama-3.1-Tulu-3-8B**[3], **Llama-3.1-Storm-8B**[4], and **Llama-3.1-SuperNova-Lite**[5] as expert models.
> >   - The performance of the models was evaluated across 7 benchmarks spanning multiple domains, including **general**, **instruction following**, **mathematics**, and **coding**. For comparison, we also included models generated using the Task Arithmetic (TA) method.
> >   - The model merged using CMM **outperforms** all expert models and the task arithmetic baseline in terms of average score across the seven benchmarks.
> >   | Model                             | MMLU  | IFEval | ARC-C | DROP  | BBH   | GSM8K | Math  | Avg   |
> >|----------------------------------|-------|--------|-------|-------|-------|-------|--------|--------|
> >| Meta-Llama-3.1-8B-Instruct       | 69.35 | 68.11  | 81.69 | 82.33 | 57.61 | 84.15 | 39.76 | 69.00 |
> >| Hermes-3-Llama-3.1-8B            | 64.29 | 59.59  | 82.03 | 77.73 | 62.25 | 81.50 | 25.20 | 64.66 |
> >| Llama-3.1-Tulu-3-8B_hf           | 66.26 | 75.18  | 66.10 | 76.68 | 61.18 | 87.87 | 40.02 | 67.61 |
> >| Llama-3.1-Storm-8B               | 68.87 | 72.18  | 81.69 | 79.01 | 56.01 | 82.94 | 35.60 | 68.04 |
> >| Llama-3.1-SuperNova-Lite         | 69.29 | 69.90  | 84.07 | 81.84 | 60.10 | 83.40 | 36.58 | 69.31 |
> >| TA                               | 67.96 | 67.03  | 82.03 | 80.02 | 69.80 | 81.43 | 41.10 | 69.91 |
> >| CMM                              | 69.81 | 70.26  | 81.69 | 83.07 | 61.11 | 84.61 | 40.96 | **70.22** |
> >
> >[1] Team Llama et al. " The llama 3 herd of models."
> >
> >[2] Teknium et al. "Hermes 3 Technical Report."
> >
> >[3] Nathan et al. "Tulu 3: Pushing Frontiers in Open Language Model Post-Training."
> >
> >[4] Ashvini Kumar Jindal. "Llama-3.1-Storm-8B: Improved SLM with Self-Curation + Model Merging."
> >
> >[5] Lucas et al. "Arcee-SuperNova: Training Pipeline and Model Composition."
>
> **Scalability**: The paper should discuss the potential scalability of the sequential CMM approach. As the number of expert models grows into the dozens or hundreds, the linear, one-by-one merging process could become a computational bottleneck, and potential "catastrophic forgetting" of the earliest merged models might become more pronounced.
> >Thank you for raising this important point regarding the scalability of the sequential merging process in CMM.
> > - We would first like to clarify that the performance degradation when merging a large number of expert models is **not specific to CMM or its sequential nature**, but rather a common challenge observed in model merging research. Some researchers have observed that the performance of even state-of-the-art model merging methods tends to **saturate after merging no more than six expert models**[1].
> > - In fact, **most existing studies on model merging typically involve only a small number of expert models**. For instance, Dai et al. merged one expert model each from the mathematics, coding, and translation domains [2], while CABS merging combined only two general-domain models [3]. To the best of our knowledge, there has not yet been a practical scenario that requires merging dozens or hundreds of large-scale expert models.
> > - We agree that scaling model merging to a large number of models raises important challenges, including potential "catastrophic forgetting" and optimization instability. We view this as a **broader open problem in the model merging domain**, not limited to sequential merging. Developing theoretical tools and practical strategies to mitigate such issues is an important direction for future work.
> >
> > [1] Wang et al. "Why Do More Experts Fail? A Theoretical Analysis of Model Merging. "
> > [2] Dai et al. "Leveraging Submodule Linearity Enhances Task Arithmetic Performance in LLMs."
> > [3] Yang et al. "CABS: Conffict-Aware and Balanced Sparsiffcation for Enhancing Model Merging. "

---

> ### Author Response · Authors · 2025-08-07
>
> Dear reviewer,
>
> We appreciate your time and effort in evaluating our work. As the discussion stage is ending soon, we wonder if our response answers your questions and addresses your concerns? Thanks again for your very constructive and insightful feedback!
>
> Best regards,
>
> Authors

---

### Official Review · Reviewer_j7J5 · 2025-07-02

**Clarity:** 2
**Significance:** 2
**Originality:** 2
**Rating:** 3
**Confidence:** 3

**Summary:**

This paper first identifies some characteristics of chemical LLMs, including the prevalence of in-house training data in the chemical domain and significant disparities in parameters and downstream functionality, which highlight the necessity of novel model merging methods. Therefore, they propose Curriculum Model Merging (CMM), a method that constructs curriculum to progressively merge the expert chemical LLMs in a moderate and continual manner. Their comprehensive experiments on two benchmark datasets show that the proposed method concentrates task-specific expertise and outperforms the state-of-the-art methods by 29.03% in terms of an overall average performance score.

**Questions:**

See weaknesses.

**Ethical Concerns:**

["NO or VERY MINOR ethics concerns only"]

**Final Justification:**

The response mostly addresses my concerns. I therefore updated my score to 3.

**Limitations:**

yes

**Quality:**

2

**Strengths And Weaknesses:**

**Strengths**

(1) This paper identifies a critical problem of merging chemical LLMs due to prevalence of in-house training data in this domain.

(2) The authors propose a model merging method to tackle this problem.

(3) Extensive experiments are conducted to demonstrate the effectiveness of the proposed method.

**Weaknesses**

(1) The experimental setting, especially those of baseline models, are not clearly described. In my understanding (with some of my best guesses), the Expert Models (e.g. Table 1) are actually evaluated by their zero-shot performance on ChemBench. The SOTA merge methods are evaluated similarly on their out-of-domain performance on ChemBench, as none of the expert models used training information from Chembench before testing. On the contrary, only the proposed CMM method gets access to Chembench information through the $S_i$ score. I would like to confirm with the authors that the understanding above is correct.

(2) If so, the comparison to all baselines in this paper seems unfair. The baselines in this paper such as TA and TIES are actually not designed for scenarios like this. These methods serve as lightweight methods with generalizability to multiple tasks, while the CMM is constructing task-specific models which are tailored to a “selected benchmark suite”.

(3) Of course, the application scenario proposed by the authors might be useful. But in such cases, the baselines to compare should not be limited to those model merging methods shown in this paper. For example, what is the performance of directly finetuning a LLM on Chembench, and what about directly using a cutting edge LLM like GPT-4o with in-context learning? In other words, this paper proposes to use model merging to obtain high performance on chemical tasks like Chembench, then it is necessary to show advantage of CMM to all kinds of machine learning methods. Only comparing CMM to model merging methods looks insufficient to me.

---

> ### Author Rebuttal · Authors · 2025-07-31
>
> We appreciate very much your constructive comments on our paper. Please kindly find our response to your comments below. We hope that our response satisfactorily addresses the issues you raised. Please feel free to let us know if you have any additional concerns or questions.
>
> **W1 & W2**: The experimental setting, especially those of baseline models, are not clearly described. The comparison to all baselines in this paper seems unfair.
> >Thank you for the insightful question regarding fairness.
> > - We use the **dev set of ChemBench** and **100 samples from Mol-Instructions** as our **validation set**, which is completely **disjoint** from the **test set**. The $S_i$ scores are computed based on the validation set. Consistent with the baseline models, the CMM-merged model does **not** have access to any information from the test set.
> > - The merged models produced by the SOTA methods get access to **the same information** as the CMM-merged model.
> >   - Our experimental process can be divided into two stages: **ranking** and **merging**. In the **ranking** stage, expert models are ranked based on their performance on the **validation set**, and corresponding weights are assigned accordingly. The **merging** stage then combines the expert models in the parameter space using different merging methods (e.g., CMM, TA, TIES, DARE_TA, SCE, AF).
> >   - The comparison is **fair and consistent** for both the SOTA merging methods and CMM, as each method receives the **same ranking information** derived from the validation set. More specifically, the TA, TIES, DARE_TA, and SCE methods also require assigning a weight to each expert model. The AF method requires specifying a merging order. In all cases, the weights or orders used are the same as those used by CMM.
>
>
> **W3**: The baselines to compare should not be limited to those model merging methods shown in this paper. In other words, this paper proposes to use model merging to obtain high performance on chemical tasks like Chembench, then it is necessary to show advantage of CMM to all kinds of machine learning methods.
> >We sincerely thank the reviewer for this insightful suggestion.
> > - **To address the point about stronger baselines**, we conducted an additional experiment where we **fine-tuned** Llama-3-8B-Instruct on the dev set of ChemBench and compared it with our CMM-merged model. Due to the limited training data, the fine-tuned model (SFT) performed poorly:
> >   | Model   | Chembench Average |
> >   | -------- | ------ |
> >   | sft model   | 19.19   |
> > - We also compare CMM with GPT-4, a leading closed-source model, based on results reported in prior work[1]:
> >   - Note that all results on ChemBench, including those for GPT-4 and our CMM-merged model, are obtained under the same **5-shot in-context learning** setting without any fine-tuning. This ensures a fair comparison across all methods.
> >   - This comparison reveals that while our CMM-merged model—constructed from **open-source** expert models without any additional training—achieves **competitive performance** on several tasks (e.g., NC, M2C, RS, TP). While GPT-4 remains stronger overall (65.89 vs. 61.43), CMM offers a compelling trade-off by delivering strong performance with **zero training cost, full reproducibility**, and **no reliance on proprietary APIs or data**.
> >   | Model  | NC    | Property_P | M2C   | C2M   | Product_P | RS    | YP    | TP    | SP    | Average |
> >   |--------|-------|------------|-------|-------|-----------|-------|-------|-------|-------|---------|
> >   | GPT-4  | 55    | 68         | 95    | 64    | 88        | 72    | 45    | 59    | 47    | 65.89   |
> >   | CMM    | 63.95 | 35.40      | 92.98 | 54.37 | 78.33     | 72.33 | 46.00 | 64.85 | 44.67 | 61.43   |
> > - Beyond its efficiency, model merging offers a key structural advantage. CMM operates in the parameter space, and is **orthogonal to the expert models themselves**—stronger expert models naturally lead to stronger merged models. This design decouples capability acquisition (via expert models) from integration (via merging), making CMM especially promising as open-weight models continue to improve.
> >
> >[1] Zhang et al. "Chemllm: A chemical large language model."

---

> ### Comment · Reviewer_j7J5 · 2025-08-05
>
> Thank you for your response, which addresses most of my concerns. I therefore raised my score.

---

> > ### Author Response · Authors · 2025-08-06
> >
> > We are pleased that the concerns raised by the reviewer have been successfully addressed. We extend our gratitude for the time and effort the reviewer dedicated to thoroughly reviewing our paper and providing valuable feedback.

---

### Official Review · Reviewer_2XdM · 2025-07-03

**Clarity:** 2
**Significance:** 3
**Originality:** 3
**Rating:** 4
**Confidence:** 3

**Summary:**

The paper proposes Curriculum Model Merging (CMM), a method for integrating multiple specialized chemical large language models (LLMs) into a unified model without requiring access to original training data or significant computational resources. CMM addresses challenges such as model disparities and imbalanced task coverage by progressively merging expert models in a curriculum-based manner. The method demonstrates superior performance, outperforming state-of-the-art merging techniques by 29.03% on average across two benchmark datasets. The paper highlights CMM's ability to generalize across predictive and generative tasks, even in expert-sparse scenarios

**Questions:**

1. Could the authors provide more theoretical analysis or intuition behind why the proposed curriculum merging order (weakest to strongest) is optimal? Are there scenarios where a different order might be more effective?
2. The paper mentions reduced computational costs, but could the authors provide concrete metrics (e.g., training time or resource usage) comparing CMM to other merging methods?
3. How might CMM perform in non-chemical domains (e.g., biology or materials science)? Are there domain-specific adjustments needed?

**Ethical Concerns:**

["NO or VERY MINOR ethics concerns only"]

**Final Justification:**

After considering the rebuttal and discussions, I find that the authors have satisfactorily addressed my main concerns. Theoretical justification for the “weakest-to-strongest” curriculum order is now supported by SAR-based analysis and concrete examples; the computation of normalized scores is clarified; and additional experiments with larger, more diverse expert sets strengthen the scalability claim. Some limitations remain—performance drop with >10 experts and lack of quantitative efficiency comparison with training-based methods—but these have limited impact on the core contributions. Overall, the rebuttal meaningfully improves the paper’s clarity and technical soundness, justifying my increased score.

**Limitations:**

Yes

**Quality:**

3

**Strengths And Weaknesses:**

The paper proposes Curriculum Model Merging (CMM), a method for integrating multiple specialized chemical large language models (LLMs) into a unified model without requiring access to original training data or significant computational resources. CMM addresses challenges such as model disparities and imbalanced task coverage by progressively merging expert models in a curriculum-based manner. The method demonstrates superior performance, outperforming state-of-the-art merging techniques by 29.03% on average across two benchmark datasets. The paper highlights CMM's ability to generalize across predictive and generative tasks, even in expert-sparse scenariosStrengths:

1. The curriculum-based merging approach is novel, particularly in the context of chemical LLMs. This work adapts and extends ideas from NLP to a domain with unique challenges.
2. This work addresses a critical gap in the chemical domain by enabling the integration of specialized LLMs while preserving privacy and reducing computational costs. The results demonstrate substantial improvements over existing methods.
3. This work presents a well-designed and thorough experimental evaluation, validating CMM's effectiveness across diverse chemical tasks. The ablation studies provide insights into the impact of merging order and weight coefficients.

Weaknesses:

1. The theoretical foundation for the merging process could be expanded to better justify the chosen curriculum strategy and weight assignments.
2. Some technical details, such as the exact computation of normalized scores for ranking models, could be more explicitly described.
3. The paper could further discuss the scalability of CMM to larger or more diverse model sets.

---

> ### Author Rebuttal · Authors · 2025-07-31
>
> We sincerely thank the reviewer for providing valuable feedback. We detail our response below point by point. Please kindly let us know whether you have any further concerns.
>
> **W1 & Q1**: Theoretical analysis for the chosen curriculum strategy and weight assignments. Could the authors provide more theoretical analysis or intuition behind why the proposed curriculum merging order (weakest to strongest) is optimal?
> >We appreciated for your concern regarding the limited theoretical analysis of curriculum model merging strategy. We provide further theoretical analysis below. We use the **Subspace Alignment Ratio (SAR)**[1] to investigate how the expert models are integrated into the merged model during the merging process.
>
> >1.
> >  The SAR is used to quantify the alignment between the subspaces of two task matrices. SAR is defined as:
>    $$
> \mathrm{SAR}(\Delta_t, \Delta_M; k_M) = \frac{ \left\| \Pi_{k_M, M} \Delta_t \right\|_F }{ \left\| \Delta_t \right\|_F },
> $$
>
> >    where $\Delta_t$ denotes a task vector, $\Delta_M$ is the difference between the parameters of the merged model and those of the base model, and $\Pi_{k_M, M}$ is the projection matrix onto the subspace spanned by the top $k_M$ left-singular vectors of $\Delta_M$. The number of singular vectors used ($k_M$) is formulated as:
> $$
> k_M = \min \{ k : \frac{ \sum_{i=k+1}^{r} \sigma_i^2 }{ \sum_{i=1}^{r} \sigma_i^2 } \leq \epsilon^2 \}
> $$
>
> >  where $\sigma_i$ denotes the singular values of $\Delta_M$, and $\epsilon$ = 0.05.
>
> 2.
> >  **A higher SAR value indicates a greater overlap between the subspaces, meaning that the merged model better inherits the capabilities of the expert model.** We observe that as more expert models are merged later in the process, the SAR values of those merged earlier tend to slightly decrease. Additionally, higher assigned weights are generally associated with higher SAR values. This suggests that placing better-performing expert models later in the merging order, and assigning them higher weights, leads to higher SAR values.
>
> 3.
> >  For better intuition, we present an example below. The table shows the evolution of the SAR value for **Molinst-molecule-8B** during the CMM merging process. This expert model was merged at **step 3**, where its SAR value increased significantly. At **step 4**, the SAR value slightly decreased due to the merging of KALE-LM-Chem-1.5. When we reduced the weight assigned to Molinst-molecule-8B at step 3, we observed a corresponding decrease in its SAR value.
> >
> >   | Metric   | Step1 | Step2 | Step3 | Step4 | Step3（reduced weight） |
> >   | -------- | ------ | ------ | ------ | ----- | ----- |
> >   | SAR   | 0.042   | 0.042   | 0.650   | 0.647  | 0.506 |
> >
> >[1] Marczak et al. "No Task Left Behind: Isotropic Model Merging with Common and Task-Specific Subspaces."
>
>
> **W2**: Some technical details, such as the exact computation of normalized scores for ranking models, could be more explicitly described.
> >Thank you for this question.
> > - We have clarified the computation details of normalized scores in **Section 4.2**. *“To enable a unified comparison of model performance across the two benchmarks, we normalize the round-trip accuracy scores from Mol-Instructions by rescaling them to a 0–100 range.”*
> > - This normalization is necessary because ChemBench scores are already reported on a 0–100 scale, whereas the Mol-Instructions round-trip accuracy scores are originally on a 0–1 scale. To ensure **fair averaging across benchmarks**, we rescale the Mol-Instructions scores to the 0–100 range, enabling a consistent comparison with ChemBench.
>
>
> **W3 & Q3**: The paper could further discuss the scalability of CMM to larger or more diverse model sets.
> >We thank this insightful question.
> > - We present an additional experiment using **a larger and more diverse collection of expert models**.
> >   - To evaluate the scalability of CMM, we extended our original experiment by introducing two additional tasks—**reasoning** and **math**—evaluated on **DROP**[1] and **GSM8K**[2], respectively. We also included two high-performing expert models on these tasks: **SuperNova-8B**[3] and **Llama-Tulu-3-8B**[4]. For comparison, we also included models generated using the Task Arithmetic (TA) method.
> >   - The performance of the six expert models and the CMM-merged model across 13 tasks is shown in the table below. The CMM-merged model **outperforms** all expert models as well as the TA-merged model in terms of overall performance. This demonstrates the effectiveness of CMM when applied to larger and more diverse model sets.
> >
> > | Model | Chembench Average | Mol-Instructions Average | DROP | GSM8K | Overall Average |
> > | :-: | :-: | :-: | :-: | :-: | :-: |
> > | Llama-3-8B-Instruct | 45.93 | 0.07 | 18.63 | 79.38 | 40.41 |
> > | GeLLMO-P6 | 31.87 | 0.08 | 81.91 | 84.53 | 36.10 |
> > | Meerkat-8b-v1.0 | 43.65 | 0 | 74.3 | 44.81 | 39.38 |
> > | Molinst-molecule-8b  | 42.97 | 0.37 | 24.15 | 62.62 | 42.12 |
> > | SuperNova-8B  | 52.5 | 0 | 81.84 | 83.40 | 49.06 |
> > | Llama-Tulu-3-8B  | 53.28 | 0 | 76.68 | 87.87 | 49.54 |
> > | KALE-LM-Chem-1.5  | 57.07  | 0.07 | 56.72 | 67.63 | 50.15 |
> > | TA  | 27.35  | 0.15 | 62.50 | 66.19 | 31.14 |
> > | CMM  | 63.80  | 0.19 | 67.84 | 79.15 | 58.40 |
> >
> > - When merging a large number of expert models (e.g., more than 10), we observe that performance tends to degrade. However, we would like to clarify that this phenomenon is **not specific to CMM**, but rather a **common limitation** across model merging methods. Prior research has shown that even state-of-the-art merging methods often experience **performance saturation after merging no more than six expert models** [5]. This highlights an open challenge in the field and calls for further theoretical understanding and methodological advances to improve the scalability of model merging.
> >
> >[1] Dheeru et al. "Drop: A reading comprehension benchmark requiring discrete reasoning over paragraphs."
> >
> >[2] Karl et al. "Training veriffers to solve math word problems."
> >
> >[3] Lucas et al. "Arcee-SuperNova: Training Pipeline and Model Composition."
> >
> >[4] Nathan et al. "Tulu 3: Pushing Frontiers in Open Language Model Post-Training."
> >
> >[5] Wang et al. "Why Do More Experts Fail? A Theoretical Analysis of Model Merging. "
>
>
> **Q2**: The paper mentions reduced computational costs, but could the authors provide concrete metrics (e.g., training time or resource usage) comparing CMM to other merging methods?
> >Thank you for this question.
> > - CMM, as well as all other merging methods in baselines are all **training-free**. The merging process takes only **a few minutes**.
> >   - The *“without computational costs”* mentioned in the paper refers to the comparison between model merging and other training-based methods such as knowledge distillation and supervised fine-tuning (SFT). Model merging integrates only homogeneous models and does not require any training or data, as it simply involves combining model parameters.

---

> > ### Comment · Reviewer_2XdM · 2025-08-08
> >
> > Thank you for the detailed rebuttal. You have addressed most of my concerns, so I will increase my score from 3 to 4.

---

> > > ### Author Response · Authors · 2025-08-09
> > >
> > > We are pleased that the concerns raised by the reviewer have been successfully addressed. We extend our gratitude for the time and effort the reviewer dedicated to thoroughly reviewing our paper and providing valuable feedback.

---

> ### Author Response · Authors · 2025-08-07
>
> Dear reviewer,
>
> We appreciate your time and effort in evaluating our work. As the discussion stage is ending soon, we wonder if our response answers your questions and addresses your concerns? Thanks again for your very constructive and insightful feedback!
>
> Best regards,
>
> Authors

---

### Author Response · Authors · 2025-08-02
**Summary**

**Summary**: In order to provide greater clarity on the revisions made to our paper and the experiments we conducted to address the reviewers' questions, we have summarized the modifications and experiments made during the rebuttal period as follows.
**Additional Experiments**:
- Scalability of CMM
  - CMM to larger and more diverse model sets. (Reviewer 2XdM W3 & Q3)
  - CMM to non-chemical domains. (Reviewer 1evb Significance) (Reviewer Chme W1)
- SFT on the dev set of ChemBench. (Reviewer j7J5 W3)
- Elucidating the choice of Merging Weight Hyperparameters. (Reviewer 1evb Q4)
- Justifing the contribution of weaker expert models. (Reviewer Chme Q2)

**Clarification**:
- Theoretical analysis for curriculum merging strategy. (Reviewer 2XdM W1 & Q1)
- An explicitly description about exact computation of normalized scores. (Reviewer 2XdM W2)
- Theoretical analysis of CMM’s scalability in the number of expert models. (Reviewer 2XdM W3)(Reviewer 1evb Scalability)
- A detailed clarification of “without computational costs”. (Reviewer 2XdM Q2)
- Experimental setting of baselines. (Reviewer j7J5 W1 &W2)
- Comparison of CMM to other kinds of machine learning methods. (Reviewer j7J5 W3)
- Limitation of CMM to Architecturally Diverse Models. (Reviewer 1evb Q1)
- Concrete Evidence for Synergistic Knowledge Generalization. (Reviewer 1evb Q2)
- Theoretical analysis of the robustness of the Single-Metric Ranking Curriculum. (Reviewer 1evb Q3)
- An explicitly description of some notations (Reviewer Chme Q1)
- Trade-off between preserving earlier capabilities and acquiring new ones. (Reviewer Chme Q3)
- Add discussion of the limitations. (Reviewer Chme Limitations)

**Writting Errors**:
- We have corrected the typos. (Reviewer Chme Q4)

We sincerely hope that our response have addressed all concerns raised by the reviewers. Please let us know if our response satisfactorily answers the questions you had for this paper. Thank you once again for your time and effort.

---

### Comment · Area_Chair_Ts4X · 2025-08-03
**Request for Response to Author Rebuttal**

Dear Reviewers,

Thank you for the thorough reviews and valuable feedback you have provided. The authors have now submitted their rebuttal, and we kindly ask that you take a moment to review their response. Please let us know whether the rebuttal adequately addresses your concerns and if it changes your overall assessment or confidence in your review.

Your input at this stage is essential for ensuring a fair and informed decision.

---

### Note · Authors · 2025-08-14

Dear Area Chairs and Reviewers,

Thank you for dedicating your time to review our paper. To facilitate your final assessment, we offer a concise summary of our discussions.

- We are encouraged by the broad consensus in favor of acceptance. Although Reviewer j7J5’s final stance remains somewhat unclear due to the unobserved updated rating, Reviewer j7J5 explicitly stated that our responses had addressed his/her concerns.
- In the initial review, our work was recognized for
   - *identifying a critical research gap* in the chemical domain by enabling the integration of specialized LLMs while preserving privacy and reducing computational costs, and
   - *proposing a novel, insightful, yet simple curriculum-based merging strategy* tailored to chemical LLM challenges, supported by
   - *well-designed, extensive experiments*, and
   - *clear, well-organized writing*.
- The reviewers suggested strengthening the work with
   - additional experiments, including
     - scalability and generalizability of CMM, which we investigated and concluded that CMM applies to **larger and more diverse** model sets and is **not limited to chemical domains**,
     - comparison with direct fine-tuning on limited data, which we conducted and concluded that CMM **not only** outperforms SFT **but also** enjoys greater efficiency,
     - justification of the synergistic effect from weaker expert models, which we provided by excluding weaker expert models and observing a performance drop,
     - sensitivity analyses of merging weight hyperparameters, which we conducted and elucidated the rationale behind our chosen values.
   - theoretical analysis for the "weakest-to-strongest" curriculum merging order, for which we leveraged **Subspace Alignment Ratio (SAR)** to illustrate that placing better-performing expert models later leads to higher SAR values of them (i.e., more contributions to the merged model).


- Beyond the above additions, we also
   - clarified our fair baseline setup, where  each method receives the **same** ranking information derived from the **validation set**,
   - proved our state-of-the-art results compared against even the leading closed-source LLM,
   - provided further details on the robustness of the ranking strategy, and
   - discussed both limitations and broader applications of CMM.

We are grateful for the constructive feedback, which has strengthened our paper and will be incorporated in the final manuscript.

Best regards,

Authors

---

### Decision · Program_Chairs · 2025-09-17

**Decision:**

Accept (poster)

**Comment:**

This paper proposes Curriculum Model Merging (CMM), a training-free framework that sequentially merges specialized chemical LLMs from weakest to strongest using a curriculum with weighted task vectors. The goal is to consolidate domain expertise without access to original training data and with low computational overhead.

Strength including:
- A simple, well motivated curriculum strategy tailored to challenges of chemical LLMs.
- Strong empirical results across predictive and generative tasks, with careful ablations on merge order and weights.
- Practical value: training free, privacy preserving, reproducible, and orthogonal to the choice of expert models.
- The Rebuttal additions, including SAR based analysis, added comparisons to SFT on limited data, and scalability checks,  strengthened the paper

The primary concerns including:
- Compute efficiency claims are still qualitative. The paper states that merging takes minutes, but lacks concrete wall clock time, memory footprint, and hardware details relative to both training based baselines and other merging methods.
- The method, by design, is limited to homogeneous architectures. The rebuttal explains this and contrasts with fusion, but the limitation should be made prominent.
- A single aggregated score may underweight niche capabilities. The rebuttal offers mitigation ideas but there is no experiment that explicitly protects rare skills during ranking.
- Authors note degradation when merging many experts, and cite prior work. More explicit guidance or safeguards would help.

Three reviewers lean toward acceptance (scores of 5, 4, and 4), while one maintains a borderline reject. The rebuttal and discussion were active and constructive. The contributions are practical and relevant to a growing area in domain specific LLMs. Remaining issues are addressable in the camera ready.